# Constitutively active receptor ADGRA3 signaling induces adipose thermogenesis

Zewei Zhao[1], Longyun Hu[1], Bigui Song[1], Tao Jiang[1], Qian Wu[1], Jiejing Lin[1], Xiaoxiao Li[1], Yi Cai[1], Jin Li[1], Bingxiu Qian[1], Siqi Liu[1], Jilu Lang[2]*, Zhonghan Yang[1]*

[1]Shenzhen Key Laboratory of Systems Medicine for inflammatory diseases, School of Medicine, Shenzhen Campus of Sun Yat-Sen University, Sun Yat-Sen University, Shenzhen, China; [2]Department of Cardiovascular Surgery, Peking University Shenzhen Hospital, Shenzhen, China

## eLife Assessment

The study highlights adhesion G-protein-coupled receptor A3 (ADGRA3) as a potential target for activating adaptive thermogenesis in both white and brown adipose tissue. This finding offers **valuable** insights for researchers in the field of adipose tissue biology and metabolism. The authors have presented additional evidence to address the reviewers' comments, including experiments conducted on primary stromal vascular fractions from adipose tissues. However, the revised manuscript fails to address several reviewer concerns, such as the measurement of whole-body energy expenditure through indirect calorimetry and the assessment of food intake. Furthermore, the nanoparticle-mediated knockdown of Adgra3 did not adequately address the tissue selectivity of ADGRA in mice. As a result, the primary claims of the study are only partially supported by the available data, leading to the conclusion that the research is deemed **incomplete**.

*For correspondence:
langjl1983@126.com (JL);
yangzhh@mail.sysu.edu.cn (ZY)

**Abstract** The induction of adipose thermogenesis plays a critical role in maintaining body temperature and improving metabolic homeostasis to combat obesity. β3-adrenoceptor (β3-AR) is widely recognized as a canonical β-adrenergic G-protein-coupled receptor (GPCR) that plays a crucial role in mediating adipose thermogenesis in mice. Nonetheless, the limited expression of β3-AR in human adipocytes restricts its clinical application. The objective of this study was to identify a GPCR that is highly expressed in human adipocytes and to explore its potential involvement in adipose thermogenesis. Our research findings have demonstrated that the adhesion G-protein-coupled receptor A3 (ADGRA3), an orphan GPCR, plays a significant role in adipose thermogenesis through its constitutively active effects. ADGRA3 exhibited high expression levels in human adipocytes and mouse brown fat. Furthermore, the knockdown of *Adgra3* resulted in an exacerbated obese phenotype and a reduction in the expression of thermogenic markers in mice. Conversely, *Adgra3* overexpression activated the adipose thermogenic program and improved metabolic homeostasis in mice without exogenous ligand. We found that ADGRA3 facilitates the biogenesis of beige human or mouse adipocytes in vitro. Moreover, hesperetin was identified as a potential agonist of ADGRA3, capable of inducing adipocyte browning and ameliorating insulin resistance in mice. In conclusion, our study demonstrated that the overexpression of constitutively active ADGRA3 or the activation of ADGRA3 by hesperetin can induce adipocyte browning by Gs-PKA-CREB axis. These findings indicate that the utilization of hesperetin and the selective overexpression of ADGRA3 in adipose tissue could serve as promising therapeutic strategies in the fight against obesity.

## Introduction

Since 1975, there has been a substantial increase in the global prevalence of obesity, with the magnitude nearly tripling. The World Health Organization projects that the prevalence of obesity among adults will exceed 20% by the year 2025 (*Organization WH, 2021*). Currently, the management of excessive adiposity poses a paramount economic burden and healthcare predicament (*Chu et al., 2018*; *Zhang et al., 2020*). In addition to the detrimental social and psychological implications, a multitude of studies have consistently demonstrated a significant association between obesity and an increased vulnerability to a range of health conditions, such as type 2 diabetes, cardiovascular diseases, and cancer (*Twig et al., 2016*; *Kim et al., 2016*; *Hyppönen et al., 2019*; *Calle, 2007*).

Activating and maintaining the thermogenesis of beige/brown fat has been shown to be effective in treating obesity and related metabolic disorders in humans (*Yoneshiro et al., 2013*; *Finlin et al., 2020*). As a well-established β-adrenergic GPCR, the β3-AR has been identified as a prominent target for stimulating adipose thermogenesis in mice. Regrettably, the clinical application of β3-AR has been impeded due to its low expression in human adipocytes and the cardiovascular risks associated with other adrenergic receptors (*Sui et al., 2019*; *Blondin et al., 2020*). G-protein-coupled receptors (GPCRs) are the most prevalent class of drug targets among all drugs approved by the U.S. Food and Drug Administration (FDA). They also play a crucial role in the clinical treatment of obesity (*Hauser et al., 2017*; *Müller et al., 2022*; *Alba et al., 2021*). Therefore, it is of clinical significance to identify novel GPCR targets that induce adipose thermogenesis.

ADGRA3 is classified as an orphan adhesion G-protein-coupled receptor (aGPCR) and exhibits the typical domains found in aGPCRs within its N-terminal extracellular region (ECR), including a leucine-rich repeat (LRR), an immunoglobulin-like domain (Ig), a hormone-binding domain (HBD), and a GAIN domain (*Nybo et al., 2023*). ADGRA3 was initially discovered as a distinctive indicator of various spermatogonial progenitor cells (*Spiess et al., 2019*; *Seandel et al., 2007*). Recent studies have shown that the orphan status of receptors has posed challenges to the study of aGPCRs. However, these studies have also uncovered a conservative mechanism of aGPCR activation, which involves the use of tethered ligands in the GAIN domain (*Vizurraga et al., 2020*; *Hamann et al., 2015*). ADGRA3 has been previously identified as a receptor capable of auto-cleavage (*Sakurai et al., 2022*). However, the functional activity of ADGRA3 in a constructive manner is still uncertain. A genome-wide association study (GWAS) demonstrated a significant correlation between single nucleotide polymorphisms (SNPs) of ADGRA3 and body weight in chickens (*Cha et al., 2021*).

Nevertheless, the precise role of ADGRA3 in the progression of obesity and adipose thermogenesis remains uncertain. This study aimed to investigate three main aspects: (*Organization WH, 2021*) the impact of ADGRA3 on browning of white adipose tissue (WAT) and brown adipose tissue (BAT), (*Chu et al., 2018*) the effects of ADGRA3 on metabolic homeostasis, and (*Zhang et al., 2020*) the underlying mechanisms by which ADGRA3 induces adipose thermogenesis.

## Results

### ADGRA3 is identified as a potential GPCR inducing the development of beige fat

We conducted a comprehensive analysis of three datasets to identify ADGRA3 as a potential GPCR target that promotes the development of beige fat (*Figure 1A*). To identify novel GPCRs that induce the biogenesis of beige fat, we conducted differential gene expression analysis (*Figure 1B*) and Venn diagram analysis (*Figure 1C*) using the GSE118849 dataset obtained from the Gene Expression Omnibus (GEO) database. Additionally, we utilized the human subcutaneous adipocytes dataset (*Figure 1C*, red) and human visceral adipocytes dataset (*Figure 1C*, purple) from the human protein atlas database to obtain genes that are highly expressed in human white adipocytes. The GSE118849 dataset comprises samples of brown adipose tissue (BAT) and inguinal white adipose tissue (iWAT) obtained from mice subjected to a 72 hr cold exposure at a temperature of 4°C.

A total of 1134 differentially expressed genes (DEGs) that exhibited up-regulation in BAT compared to iWAT under cold stimulation were identified in the analysis, which might play a role in adipose thermogenesis. These DEGs were further screened to identify highly expressed GPCRs in BAT relative to WAT (*Figure 1B*, red). We conducted additional annotation on 1134 DEGs and identified that 27 of these genes were associated with the encoding of GPCRs (*Supplementary file 2*). Among the set

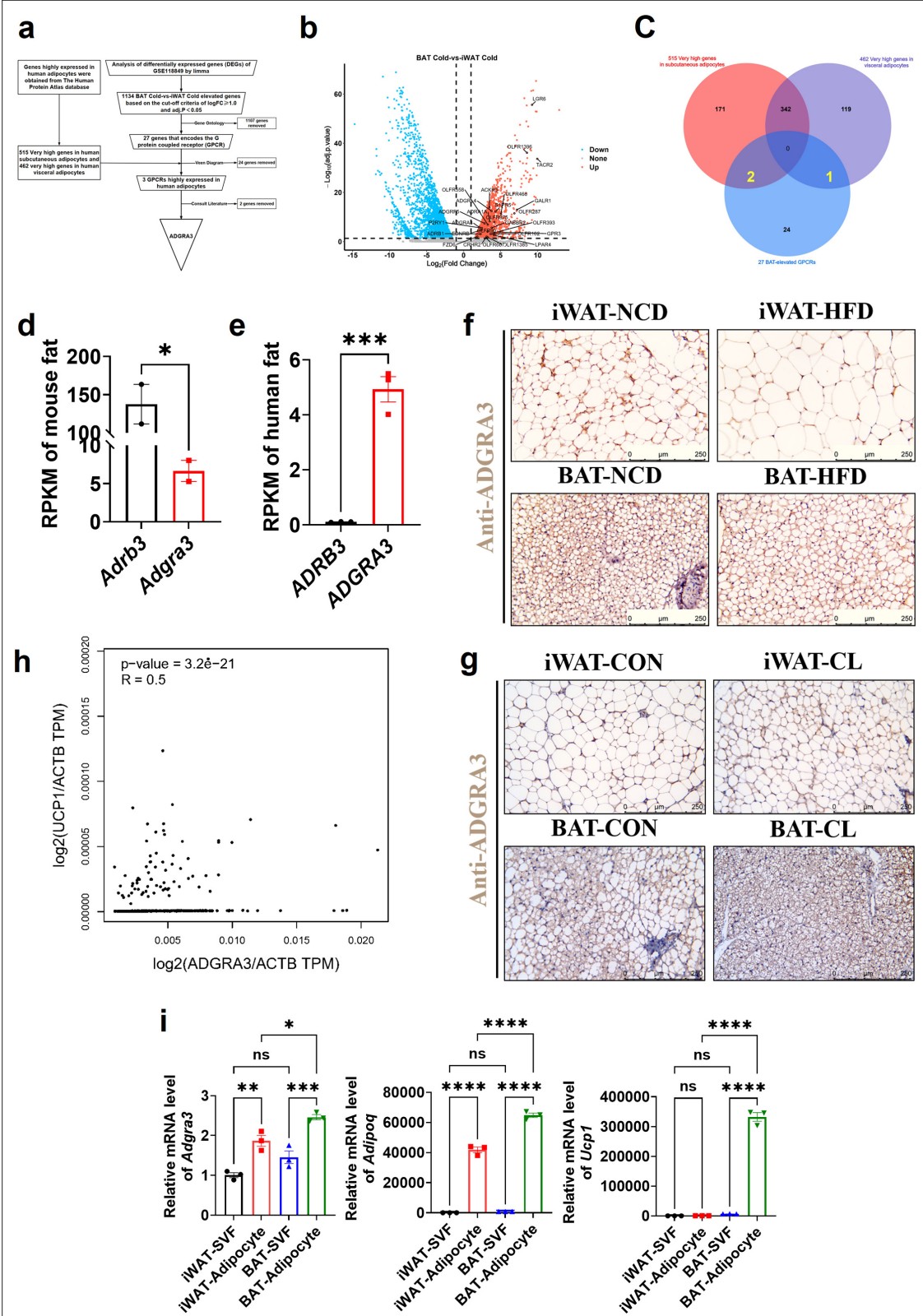

**Figure 1.** ADGRA3 is a high-expressed GPCR in human adipocytes and mouse brown fat. (**A–F**) ADGRA3 screening as a high-expressed GPCR in human adipocytes and mouse brown fat via comprehensive analysis. Brown adipose tissue and subcutaneous WAT were dissected from mice that were treated in cold (4°C) temperature for 72 hr. A total of six samples with three replicates for each adipose tissue were evaluated. The datasets of human subcutaneous adipocytes and human visceral adipocytes were acquired from the human protein atlas database. (**A**) Flowchart of screening. (**B**) Volcano

*Figure 1 continued on next page*

*Figure 1 continued*

plot summarizing the differentially expressed genes (DEGs) between cold temperature BAT group and cold temperature iWAT group. Blue and red shading are used to indicate down-regulation and up-regulation, respectively. (**C**) 27 BAT-elevated GPCRs from transcriptome, 515 very high genes in subcutaneous adipocytes and 462 very high genes in visceral adipocytes from the human protein atlas database were analyzed by using a Venn diagram. (**D–E**) The RPKM of *ADRB3* and *ADGRA3* genes in mouse fat (**D**) from Mouse ENCODE transcriptome data (PRJNA66167, N=2) and human fat (**E**) from HPA RNA-seq normal tissues (PRJEB4337, N=3). (**F**) C57BL/6 J mice fed with a NCD or a HFD for 12 weeks. Representative images of iWAT and BAT stained with ADGRA3. Scale bars, 250 μm. (**G**) C57BL/6 J mice fed with a HFD for 12 weeks were injected with vehicle or CL (1 mg/kg daily) over 7 days. Representative images of iWAT and BAT stained with ADGRA3. Scale bars, 250 μm. (**H**) Correlation between *UCP1* expression level normalized by *ACTB* gene and *ADGRA3* expression level normalized by *ACTB* gene in human subcutaneous fat dataset from GTEx Portal database (N=663). (**I**) qPCR analysis of *Adgra3, Adipoq* and *Ucp1* genes in Stromal Vascular Fraction (SVF) and mature adipocyte isolated from iWAT and BAT (N=3 for each group). iWAT, inguinal white adipose tissue; BAT, brown adipose tissue; RPKM, Reads Per Kilobase per Million mapped reads; TPM, Transcripts Per Kilobase Million; GPCR, G-protein-coupled receptor; NCD, normal chow diet; HFD, high-fat diet; CL, CL-316,243; SVF, Stromal Vascular Fraction. All data are presented as mean ± *SEM*. Statistical significance was determined by unpaired two-tailed student's t-test (**D–E**), simple linear regression (**H**) and one-way ANOVA (**I**).

The online version of this article includes the following source data and figure supplement(s) for figure 1:

**Source data 1.** Numerical source data for *Figure 1*.

**Figure supplement 1.** ADGRA3 positively correlated with beige fat.

**Figure supplement 1—source data 1.** Numerical source data for *Figure 1—figure supplement 1*.

of 27 genes, it was found that 24 genes were not present in the group of genes that exhibited high expression levels in human adipocytes, as determined by the human protein atlas database. Consequently, these 24 genes were excluded from further analysis. We conducted a comprehensive literature review and discovered that out of the three remaining GPCRs namely ADGRA3, ADRA1A, and ADRB1, only ADGRA3 has not been documented to have any association with brown fat. Therefore, our research subsequently shifted towards investigating the potential regulatory role of ADGRA3 in obesity and brown fat.

The findings indicated that the level of *Adgra3* expression in mouse adipose tissue (*Figure 1D*) was comparatively lower than that of *Adrb3*, the coding gene for β3-AR. Conversely, in human adipose tissue, *ADGRA3* expression was observed to be higher than that of *ADRB3* (*Figure 1E and F*, *Figure 1—figure supplement 1E*). We conducted an investigation to examine the regulatory effects of a high-fat diet on the transcription of *Adgra3* and *Ucp1* (Uncoupling protein 1, a functional protein and marker of beige/brown fat). The findings of the study demonstrated that a HFD had a significant inhibitory effect on the expression of ADGRA3 and UCP1 in iWAT and BAT, while CL robustly increased the expression of ADGRA3 and UCP1 in iWAT and BAT (*Figure 1F-G*, *Figure 1—figure supplement 1A-D*). Interestingly, in human subcutaneous fat, there was a moderate positive correlation between the expression level of *ADGRA3* and the expression level of *UCP1* (R=0.5, *Figure 1H*). On the other hand, the expression level of *ADRB3* showed a weak positive correlation with the expression level of *UCP1* (R=0.21, *Figure 1—figure supplement 1F*). The data presented in this study indicate that ADGRA3 is a GPCR that exhibits high expression levels in BAT and may participate in inducing adipose thermogenesis.

## Adgra3 overexpression induces the biogenesis of beige adipocytes in vitro

To ascertain the predominant expression of ADGRA3, the isolation of stromal Vascular Fraction (SVF) and mature adipocytes from WAT and BAT was conducted for subsequent validation. The results showed that ADGRA3 is predominantly expressed in adipocytes. Furthermore, the expression level of ADGRA3 in BAT adipocytes was found to be higher compared to WAT adipocytes (*Figure 1I*). However, no significant difference was observed in the expression level of ADGRA3 in the SVF of WAT and BAT (*Figure 1I*). Moreover, it was observed that the modulation of the expression levels of *Adgra3/ADGRA3* and *Ucp1/UCP1* exhibited a similar pattern during the differentiation process between mouse and human adipocytes (*Figure 1—figure supplement 1G–H*).

To investigate the role of ADGRA3 in the biogenesis of beige adipocytes, we conducted an experiment where we transformed pre-adipocytes 3T3-L1 into mature beige-like adipocytes with a knockdown of *Adgra3*. Our findings indicate that the knockdown of *Adgra3* resulted in a decrease in the expression of genes related to thermogenesis and lipolysis (*Figure 2A*). Western blot analysis and

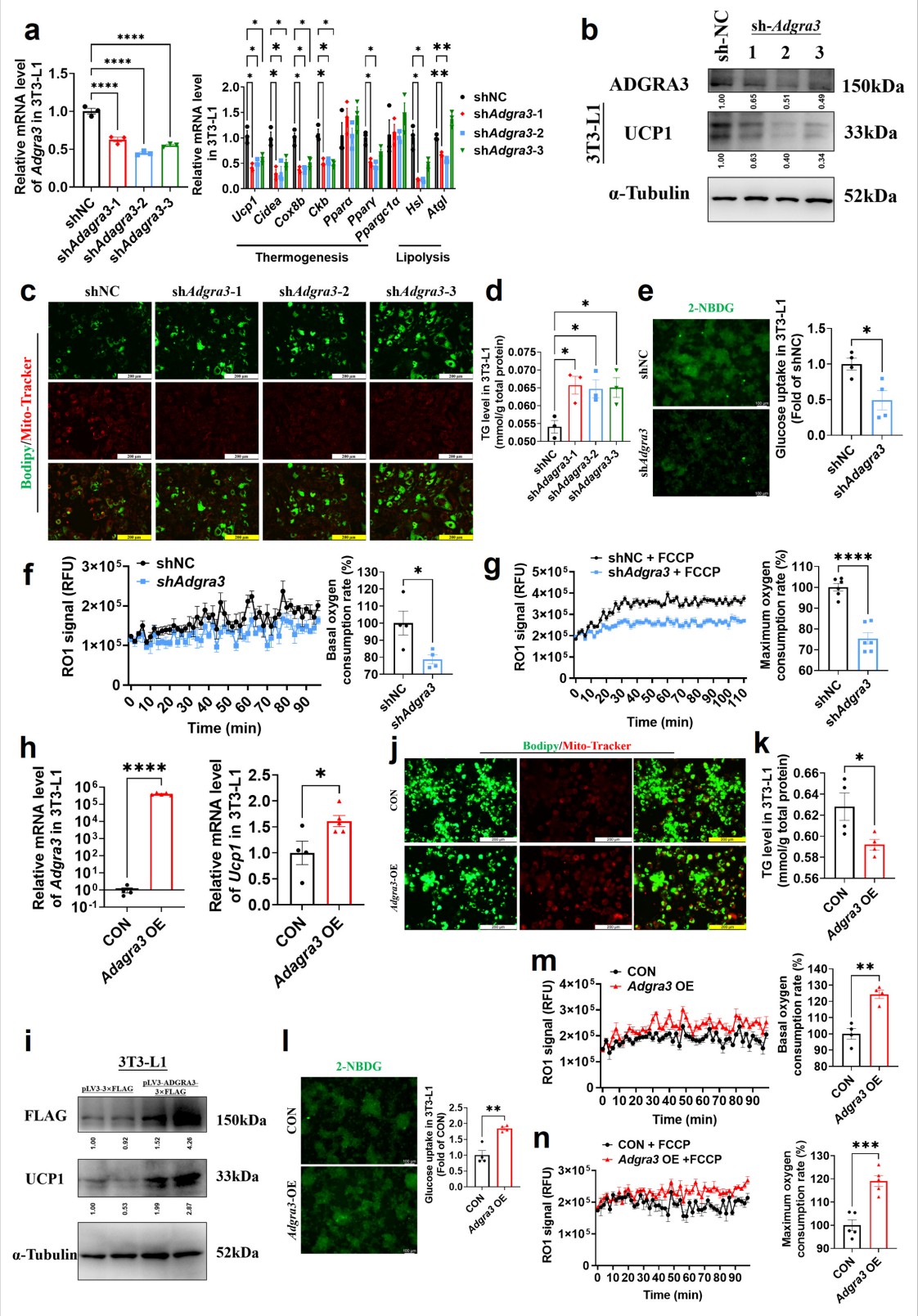

**Figure 2.** *Adgra3* overexpression promotes the biogenesis of beige adipocytes. (**A, H**) qPCR analysis of *Adgra3*, thermogenesis and lipolysis genes in 3T3-L1 mature beige-like adipocytes (A: N=3 for each group; H: N=4 for CON, N=5 for *Adgra3* OE). (**B, I**) Western blot analysis for level of ADGRA3, UCP1 and ADGRA3–3×FLAG protein in 3T3-L1 mature beige-like adipocytes treated with sh*Adgra3* (pLKO.1-U6-sh*Adgra3*-(1/2/3) plasmid encapsulated in nanomaterials), shNC (pLKO.1-U6-shNC plasmid encapsulated in nanomaterials), *Adgra3* OE (pLV3-CMV-*Adgra3*(mouse)–3×FLAG

*Figure 2 continued on next page*

*Figure 2 continued*

plasmid encapsulated in nanomaterials) or CON (pLV3-CMV-MCS-3×FLAG plasmid encapsulated in nanomaterials). The ImageJ software was used for gray scanning. (**C, J**) Bodipy green staining for lipid droplet and Mito-Tracker red staining for mitochondria in 3T3-L1 mature beige-like adipocytes. Scale bars, 200 μm. (**D, K**) The level of intracellular triglyceride in 3T3-L1 mature beige-like adipocytes (D: N=3 for each group; K: N=4 for each group). (**E, I**) Glucose uptake assay in 3T3-L1 mature beige-like adipocytes and staining intensity analysis diagram (right, N=4 for each group). (**F, M**) When 3T3-L1 mature beige-like adipocytes were treated with shNC, sh*Adgra3*, CON or *Adgra3* OE, fluorescence of the oxygen probe (RO1) in the cells was monitored and the rate of basal oxygen consumption was analyzed (N=4 for each group). (**G, N**) When FCCP-treaded 3T3-L1 mature beige-like adipocytes were treated with shNC, sh*Adgra3*, CON or *Adgra3* OE, fluorescence of the oxygen probe (RO1) in the cells was monitored and the rate of maximum oxygen consumption was analyzed (G: N=6 for each group; N: N=5 for each group). All data are presented as mean ± *SEM*. Statistical significance was determined by unpaired two-tailed student's t-test (**E–H and K–N**) and one-way ANOVA (**A and D**).

The online version of this article includes the following source data for figure 2:

**Source data 1.** Raw uncropped blots for *Figure 2*.

**Source data 2.** Uncropped and labeled blots for *Figure 2*.

**Source data 3.** Numerical source data for *Figure 2*.

Mito-Tracker staining revealed a decrease in the expression of UCP1 (*Figure 2B*) and a reduction in the number of mitochondria (*Figure 2C*) following *Adgra3* knockdown. Lipid droplet fluorescence staining and intracellular triglyceride assay were performed on adipocytes to assess the impact of *Adgra3* knockdown. The results revealed a significant increase in the number of lipid droplets and intracellular triglyceride levels (*Figure 2C–D*) following *Adgra3* knockdown. Moreover, the uptake of 2-deoxy-D-glucose (2-NBDG), a fluorescently-labeled deoxyglucose analog, by adipocytes was significantly inhibited following the knockdown of *Adgra3* (*Figure 2E*). Furthermore, oxygen consumption rate (OCR) was detected to verify the effect of ADGRA3 on the oxygen consumption of adipocytes. The results indicated that the loss of ADGRA3 decreased the both basal and max OCR of adipocytes (*Figure 2F–G*).

Following the overexpression of *Adgra3*, there was an observed up-regulation in the expression of UCP1 in 3T3-L1 mature beige-like adipocytes (*Figure 2H–I*). Additionally, Mito-Tracker staining revealed an increase in the quantity of mitochondria (*Figure 2J*). There was a notable reduction observed in the lipid droplets and intracellular triglyceride levels (*Figure 2J–K*) subsequent to the overexpression of *Adgra3*. Moreover, the findings indicated that the overexpression of *Adgra3* resulted in an increased uptake of 2-NBDG by adipocytes (*Figure 2L*) and increased basal and maximum OCR (*Figure 2M–N*). The presented data suggest that ADGRA3 has the ability to stimulate the formation of beige adipocytes in vitro.

## Adgra3 knockdown suppresses adipose thermogenic program and impairs metabolic homeostasis in vivo

To evaluate the role of ADGRA3 in the biogenesis of beige fat in vivo, mice fed with a NCD were injected with shNC or sh*Adgra3* for 28 days (*Figure 3A*). After knocking down *Adgra3* in mice (*Figure 3— figure supplement 1A–B*), there was a significant increase in the weight of sh*Adgra3* mice (mice with *Adgra3* knockdown; *Figure 3B*). Furthermore, the food intake of sh*Adgra3* mice was elevated slightly (*Figure 3—figure supplement 1C*). Serum triacylglycerol (TG) levels (*Figure 3—figure supplement 1D*), weight of iWAT, epididymal white adipose tissue (eWAT) and BAT (*Figure 3C*) were significantly higher in sh*Adgra3* mice. Liver weight (*Figure 3C*) and TG levels in the liver (*Figure 3—figure supplement 1E*) did not show a significant difference between shNC mice and sh*Adgra3* mice. Meanwhile, hematoxylin-eosin staining showed that *Adgra3* knockdown induced adipose expansion in iWAT (*Figure 3—figure supplement 1F*), eWAT (*Figure 3—figure supplement 1G*), and BAT (*Figure 3— figure supplement 1F*) but not lead to hepatic steatosis (*Figure 3—figure supplement 1G*).

Remarkably, the knockdown of *Adgra3* resulted in a significant reduction in both body temperature (*Figure 3D*) and BAT temperature (*Figure 3E*). Given the crucial influence of thyroid activity on thermogenesis, we measured the levels of serum-free tetraiodothyronine (fT4) to evaluate the consequences of *Adgra3* knockdown on thyroid activity, which indicated that the nanoparticle-mediated *Adgra3* knockdown does not exert an inhibitory effect on thyroid activity (*Figure 3—figure supplement 1H*). The knockdown of *Adgra3* resulted in a significant decrease in the expression of genes related to thermogenesis and lipolysis in both iWAT (*Figure 3F*) and BAT (*Figure 3G*). Moreover, the western blot analysis (*Figure 3H–I*) and immunohistochemical staining (*Figure 3J*) of UCP1 revealed comparable

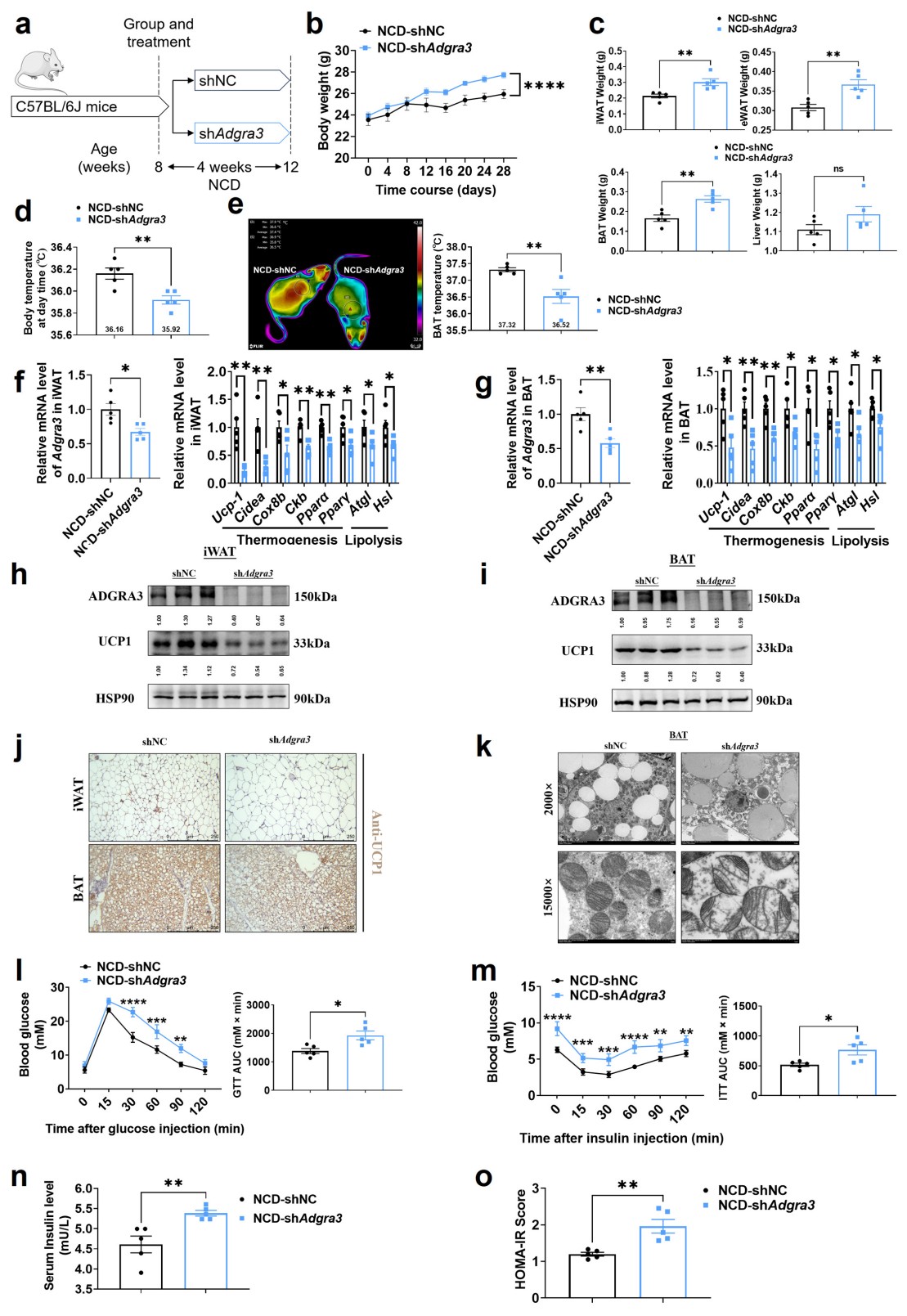

**Figure 3.** Knockdown of *Adgra3* suppressed the adipose thermogenic program and impaired metabolic homeostasis in mice. (**A**) Experimental schematic. C57BL/6 J mice fed with a NCD for eight weeks were injected with sh*Adgra3* (pLKO.1-U6-sh*Adgra3*-2 plasmid encapsulated in nanomaterials) or shNC (pLKO.1-U6-shNC plasmid encapsulated in nanomaterials) twice a week for four weeks. (**B–D**) Changes in body mass (**B**), tissue weight (**C**) and body temperature (**D**) in mice injected with shNC or sh*Adgra3* for 28 days (N=5 for each group). (**E**) Thermal image and BAT temperature

*Figure 3 continued on next page*

*Figure 3 continued*

of mice injected with shNC or sh*Adgra3* for 28 days (N=5 for each group). (**F–G**) qPCR analysis of *Adgra3*, genes associated with thermogenesis and lipolysis in iWAT (**F**) and BAT (**G**) from different treatment mice (N=5 for each group). (**H–I**) Western-blot analysis for the level of ADGRA3 and UCP1 protein in iWAT (**H**) and BAT (**I**) from differently treated mice. (**J**) Representative images of iWAT (top) and BAT (bottom) stained with UCP1. Scale bars, 250 µm. (**K**) Transmission electron microscope photograph of BAT treated with shNC or sh*Adgra3*. (**L**) Glucose tolerance test (GTT) was conducted by intraperitoneal injection of glucose (2 g/kg) and measurement of blood glucose concentration with a OneTouch Ultra Glucometer at designed time points in 6 hr fasted mice (N=5 for each group). (**M**) Insulin tolerance test (ITT) was done by intraperitoneal injection of insulin (0.5 U/kg) and measurement of blood glucose concentration by a OneTouch Ultra Glucometer at designed time points in 6 hr fasted mice (N=5 for each group). (**N–O**) The fasting serum insulin (**N**) and HOMA-IR (**O**) in mice injected with either shNC or sh*Adgra3* for 28 days (N=5 for each group). HOMA-IR=Fasting glucose level (mmol/L) * Fasting insulin level (mIU/L) /22.5. NCD, normal chow diet; iWAT, inguinal white adipose tissue; BAT, brown adipose tissue; GTT, Glucose tolerance test; ITT, Insulin tolerance test; HOMA-IR, homeostasis model assessment of insulin resistance. All data are presented as mean ± SEM. Statistical significance was determined by unpaired two-tailed student's t-test (**C–G and N–O**) and two-way ANOVA (**B and L–M**).

The online version of this article includes the following source data and figure supplement(s) for figure 3:

**Source data 1.** Raw uncropped blots for *Figure 3*.

**Source data 2.** Uncropped and labeled blots for *Figure 3*.

**Source data 3.** Numerical source data for *Figure 3*.

**Figure supplement 1.** Characterization of wild-type and *Adgra3*-knockdown mice.

**Figure supplement 1—source data 1.** Numerical source data for *Figure 3—figure supplement 1*.

**Figure supplement 2.** *ADGRA3* high expressed gene sets in human subcutaneous fat are enriched to lipid metabolism and adipocyte differentiation.

outcomes. Additionally, it was observed that the knockdown of *Adgra3* resulted in an increase in the size of lipid droplets and a decrease in the number of mitochondria in BAT (*Figure 3K*). Furthermore, nanomaterials carrying sh*Adgra3* were directly injected into BAT (*Figure 3—figure supplement 1I*), resulting in knockdown of *Adgra3* and down-regulation of thermogenic and lipolysis-related genes, as compared to BAT injected with shNC (*Figure 3—figure supplement 1J*). These findings indicate that ADGRA3 plays a crucial role as a receptor in the biogenesis of beige fat and the activation of BAT.

Moreover, the genes that were highly expressed in ADGRA3 high-expressed human subcutaneous adipose tissue (*Figure 3—figure supplement 2A*, red) exhibited enrichment in various biological processes. These processes included hyperinsulinism, obesity (*Figure 3—figure supplement 2B*), metabolic processes (*Figure 3—figure supplement 2C*), adipogenesis (*Figure 3—figure supplement 2D*), regulation of lipolysis in adipocytes (*Figure 3—figure supplement 2E*), and lipid metabolism (*Figure 3—figure supplement 2F*). GSEA was conducted to search the enriched KEGG pathways based on the expression level of ADGRA3 in human subcutaneous adipose dataset and human visceral adipose dataset from GTEx portal database. For ADGRA3 high-expressed group, both subcutaneous adipose dataset (*Figure 3—figure supplement 2G*) and visceral adipose dataset (*Figure 3—figure supplement 2H*) enriched in insulin signaling pathway, which indicates that ADGRA3 may be involved in the regulation of glucose metabolism in addition to its influence on lipid metabolism. Furthermore, it was observed that sh*Adgra3* mice exhibited significant disruptions in overall glycemic homeostasis (*Figure 3L*) and insulin sensitivity (*Figure 3M*). Moreover, the fasting serum insulin level was increased and the homeostasis model assessment of insulin resistance (HOMA-IR) showed an increase in sh*Adgra3* mice (*Figure 3N–O*). Hence, the findings of this study provide evidence that the knockdown of *Adgra3* hampers adipose thermogenesis and disrupts metabolic homeostasis in vivo.

## ADGRA3 activates the adipose thermogenic program and counteracts metabolic disease in vivo

To identify whether *Adgra3* overexpression induces adipose thermogenesis and improves the metabolic homeostasis against obesity, *Adgra3* OE and CON were injected i.p. into mice fed with a NCD (*Figure 4—figure supplement 1A*) and a HFD (*Figure 4A*), thereby establishing models of *Adgra3*-overexpressed mice (*Figure 4F-G*, *Figure 4—figure supplement 1B-C*). The growth of body weight of *Adgra3* OE mice was alleviated (*Figure 4B*) during the HFD feeding accompanied with a slight decrease of food intake (*Figure 4—figure supplement 2A*). Furthermore, the weight of iWAT, eWAT, BAT, and liver (*Figure 4C*) were significantly decreased in *Adgra3* OE mice. The *Adgra3* OE mice exhibited an elevation in both body temperature (*Figure 4D*, *Figure 4—figure supplement 1D*) and BAT temperature (*Figure 4E*, *Figure 4—figure supplement 1E*), while there was no difference

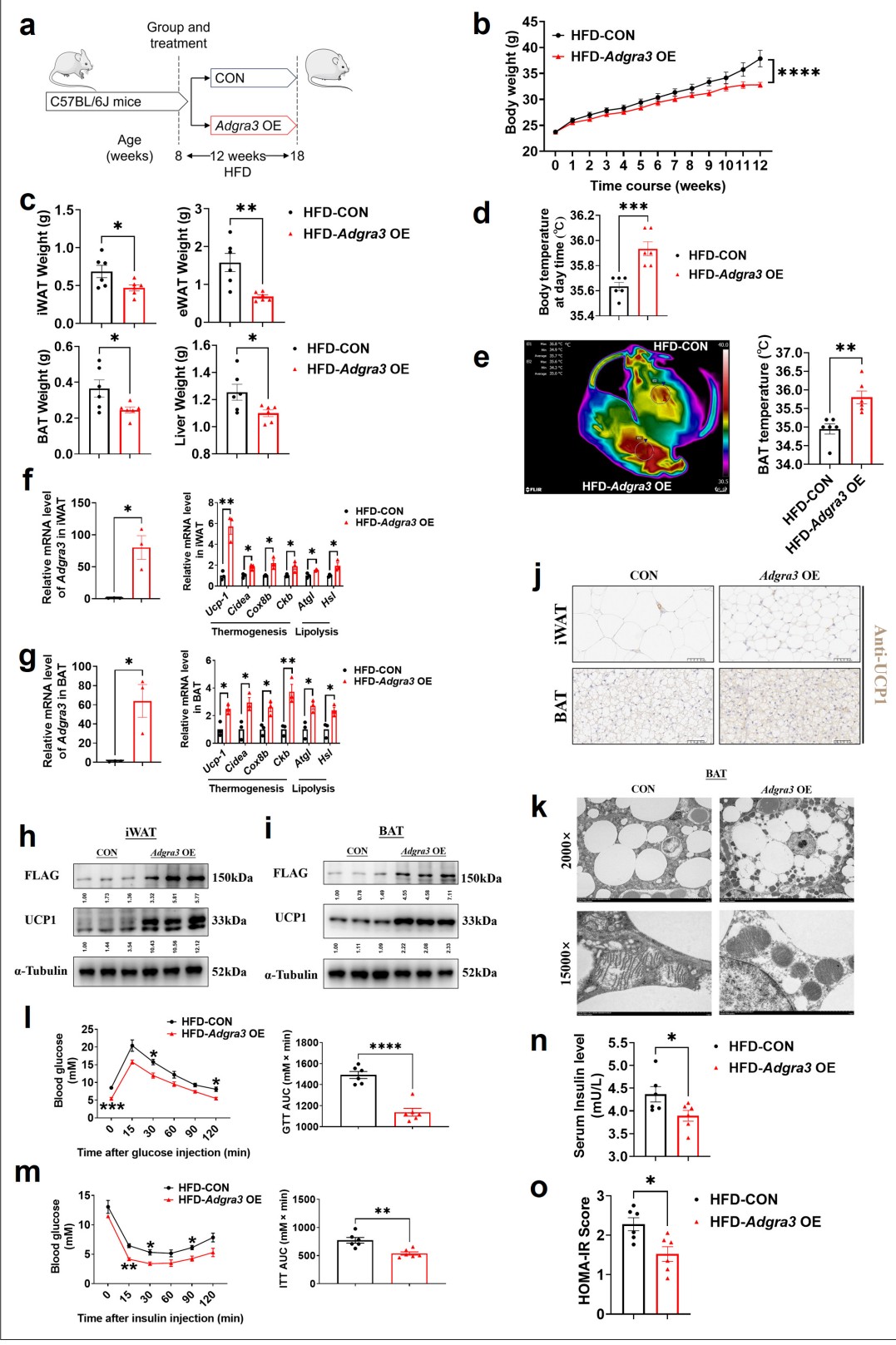

**Figure 4.** *Adgra3* overexpression activated the adipose thermogenic program and facilitated metabolic homeostasis in mice with diet-induced obesity (DIO). (**A**) Experimental schematic. C57BL/6 J mice were fed with a HFD and injected with *Adgra3* OE (pLV3-CMV-*Adgra3* (mouse)–3×FLAG plasmid encapsulated in nanomaterials) or CON (pLV3-CMV-MCS-3×FLAG plasmid encapsulated in nanomaterials) once a week for 12 weeks. (**B–**

*Figure 4 continued on next page*

*Figure 4 continued*

**D**) Changes in body mass (**B**), tissue weight (**C**) and body temperature (**D**) of mice injected with CON or *Adgra3* OE (N=6 for each group). (**E**) Thermal image and BAT temperature in mice injected with CON or *Adgra3* OE (N=6 for each group). (**F–G**) qPCR analysis of *Adgra3*, genes associated with thermogenesis and lipolysis in iWAT (**F**) and BAT (**G**) from different treatment mice (N=3 for each group). (**H–I**) Western-blot analysis for the level of ADGRA3−3×FLAG and UCP1 protein in iWAT (**H**) and BAT (**I**) from differently treated mice. (**J**) Representative images of iWAT (top; Scale bars, 50 µm.) and BAT (bottom; Scale bars, 50 µm.) stained with UCP1. (**K**) Transmission electron microscope photograph of BAT treated with CON or *Adgra3* OE. (**L**) Glucose tolerance test (GTT) was conducted by intraperitoneal injection of glucose (1 g/kg) and measurement of blood glucose concentration with a OneTouch Ultra Glucometer at designed time points in six hours fasted mice (N=6 for each group). (**M**) Insulin tolerance test (ITT) was done by intraperitoneal injection of insulin (1 U/kg) and measurement of blood glucose concentration by a OneTouch Ultra Glucometer at designed time points in six hours fasted mice (N=6 for each group). (**N–O**) The fasting serum insulin (**N**) and HOMA-IR (**O**) in mice injected with CON or *Adgra3* OE (N=6 for each group). HOMA-IR=Fasting glucose level (mmol/L) * Fasting insulin level (mIU/L) /22.5. HFD, high-fat diet; iWAT, inguinal white adipose tissue; BAT, brown adipose tissue; GTT, Glucose tolerance test; ITT, Insulin tolerance test; HOMA-IR, homeostasis model assessment of insulin resistance. All data are presented as mean ± SEM. Statistical significance was determined by unpaired two-tailed student's t-test (**C–G and N–O**) and two-way ANOVA (**B and L–M**).

The online version of this article includes the following source data and figure supplement(s) for figure 4:

**Source data 1.** Raw uncropped blots for *Figure 4*.

**Source data 2.** Uncropped and labeled blots for *Figure 4*.

**Source data 3.** Numerical source data for *Figure 4*.

**Figure supplement 1.** *Adgra3* overexpression activated the adipose thermogenic program in mice.

**Figure supplement 1—source data 1.** Raw uncropped blots for *Figure 4—figure supplement 1*.

**Figure supplement 1—source data 2.** Uncropped and labeled blots for *Figure 4—figure supplement 1*.

**Figure supplement 1—source data 3.** Numerical source data for *Figure 4—figure supplement 1*.

**Figure supplement 2.** Characterization of wild-type and *Adgra3*-overexpressed mice.

**Figure supplement 2—source data 1.** Numerical source data for *Figure 4—figure supplement 2*.

in serum fT4 levels (*Figure 4—figure supplements 1F and 2B*). Meanwhile, *Adgra3* overexpression decreased the TG level in serum and liver (*Figure 4—figure supplement 2C–D*) as well as the area of adipocytes in iWAT, eWAT, BAT, and liver (*Figure 4—figure supplements 1G and 2E-F*).

Moreover, the expression levels of thermogenic and lipolysis-related genes were elevated in iWAT (*Figure 4F*, *Figure 4—figure supplement 1H*) and BAT (*Figure 4G*, *Figure 4—figure supplement 1I*). Western blot (*Figure 4H-I*, *Figure 4—figure supplement 1J-K*) and immunohistochemical staining of UCP1 (*Figure 4J*, *Figure 4—figure supplement 1L*) showed that the expression of UCP1 was increased dramatically in iWAT and BAT after *Adgra3* overexpression. In addition, we found that after *Adgra3* overexpression, BAT presented multiple thermogenesis fat features (*Figure 4K*). Furthermore, nanomaterials carrying *Adgra3* OE were directly injected into BAT (*Figure 4—figure supplement 2G*), resulting in overexpression of *Adgra3* and upregulation of thermogenic and lipolysis-related genes, as compared to BAT injected with CON (*Figure 4—figure supplement 2H*). These findings indicate that the overexpression of *Adgra3* is capable of inducing the hallmarks of thermogenesis in mice, independently of other organs.

We then investigated the metabolic impact of ADGRA3. The glucose tolerance test (GTT) presented that *Adgra3* overexpression improved the glucose homeostasis of HFD mice (*Figure 4L*). The insulin tolerance test (ITT) showed that *Adgra3* overexpression alleviated the insulin resistance of HFD mice (*Figure 4M*). Moreover, the fasting serum insulin level was reduced after *Adgra3* overexpression (*Figure 4N*) and the HOMA-IR also showed a robust improvement (*Figure 4O*) in *Adgra3*OE mice. Taken together, *Adgra3* overexpression activates the adipose thermogenic program and improves the metabolic homeostasis in diet-induced obese mice against obesity and insulin resistance in vivo.

## ADGRA3 activates the adipose thermogenic program via the $G_s$-PKA-CREB axis

To ascertain the ADGRA3-conjugated Gα protein, we conducted an overexpression of FLAG-labeled mouse ADGRA3 and four different types of His-labeled $G_\alpha$ proteins ($G_s$, $G_i$, $G_q$ and $G_{12}$) in 293T cells. The lysate obtained from the 293T cells was then utilized for the subsequent co-immunoprecipitation (co-IP) analysis. The results of the co-IP experiment demonstrated that mouse ADGRA3 coupled to the $G_s$ protein (*Figure 5A–B*), while no interaction was observed with the other three Gα proteins ($G_i$, $G_q$ and $G_{12}$; *Figure 5—figure supplement 1A–C*) ADGRA3 exhibits intrinsic and auto-cleavable receptor activity, allowing it to signal even in the absence of an exogenous ligand (*Spiess et al., 2019*; *Sakurai et al., 2022*). Hence, the overexpression of *Adgra3* is capable of inducing cAMP production (*Figure 5C*), which serves as a second messenger indicating the activation of downstream signals mediated by $G_s$ protein. This response is comparable to the effect of a ligand. However, there is no production of IP1, which is a metabolite of the downstream second messenger IP3 associated with $G_q$ protein (*Figure 5—figure supplement 1D*). Additionally, our findings indicate that the effect of *Adgra3* overexpression on cAMP production is dependent on $G_s$ protein (*Figure 5D*, *Figure 5—figure supplement 1E*). These results suggest that ADGRA3 is involved in the coupling of $G_s$ protein, leading to the stimulation of downstream cAMP production.

Hence, it was hypothesized that the overexpression of *Adgra3* could potentially stimulate adipocyte thermogenesis by activating the PKA signaling pathway. As expected, the Western-blot analysis revealed that the overexpression of *Adgra3* leads to an elevation in the phosphorylated form of CREB (p-CREB), indicating an increase in PKA-CREB signaling activity. This effect was observed in 3T3-L1 (*Figure 5E*), as well as in the iWAT and BAT (*Figure 5F*). Consistently, the knockdown of *Adgra3* resulted in a decrease in the level of p-CREB in 3T3-L1 (*Figure 5G*), as well as in iWAT and BAT (*Figure 5H*). To investigate the potential role of *Adgra3* overexpression in promoting the biogenesis of beige adipocytes and activating the PKA-CREB signaling pathway via $G_s$ protein, we conducted an experiment using 3T3-L1 cells. The cells were treated with *Adgra3* OE and sh*Gnas*, respectively. *Adgra3* overexpression was found to be adequate in inducing the expression of UCP1 in 3T3-L1 cells. However, this effect was observed to be eliminated when *Gnas* was knocked down (*Figure 5I*, *Figure 5—figure supplement 1E*). Furthermore, the utilization of PKAi (protein kinase A inhibitor, H-89) was employed to ascertain the dependence of the browning effect of Adgra3 overexpression on the PKA-CREB signal. The results showed that PKAi effectively inhibited the activation of PKA-CREB signaling and UCP1 expression induced by *Adgra3* overexpression (*Figure 5J*, *Figure 5—figure supplement 1F*). These results suggest that the observed browning effect resulting from *Adgra3* overexpression is mediated through the PKA-CREB signaling pathway. Collectively, these findings indicate that ADGRA3 facilitates the biogenesis of beige adipocytes through the $G_s$-PKA-CREB signaling pathway.

## Hesperetin: a screened ADGRA3 agonist that induces the biogenesis of beige adipocytes

Considering the difficulty of overexpressing ADGRA3 in clinical application, hesperetin was screened as a potential agonist of ADGRA3 by PRESTO-Salsa database (*Figure 6A*). The results showed that hesperetin treatment stimulates cAMP production (*Figure 6B*) and increases the expression level of UCP1 and p-CREB (*Figure 6C–D*). To verify whether hesperetin induces the biogenesis of beige adipocyte and activates PKA-CREB signal via ADGRA3, we treated 3T3-L1 with hesperetin and sh*Adgra3*, respectively. We found that the induction effect of hesperetin on UCP1 and p-CREB is eliminated when *Adgra3* is knocked down (*Figure 6E–F*). In addition, OCR was detected to verify the effect of hesperetin on the oxygen consumption of adipocytes. The results indicated that hesperetin increased the both basal and max OCR of adipocytes, which was ADGRA3-dependent (*Figure 6G–H*).

Moreover, the results showed that the induction effect of hesperetin on UCP1 and p-CREB is attenuated after *Gnas* knocked down (*Figure 6I–J*), suggesting that hesperetin up-regulates UCP1 and activates PKA-CREB axis dependent on $G_s$. Furthermore, PKAi was used to verify whether the browning effect of hesperetin was dependent on PKA-CREB signal. The results revealed that hesperetin treatment resulted in the upregulation of UCP1 and p-CREB. However, this effect was found to be eliminated when PKAi was applied (*Figure 6K–L*), suggesting that the induction of UCP1 and p-CREB by

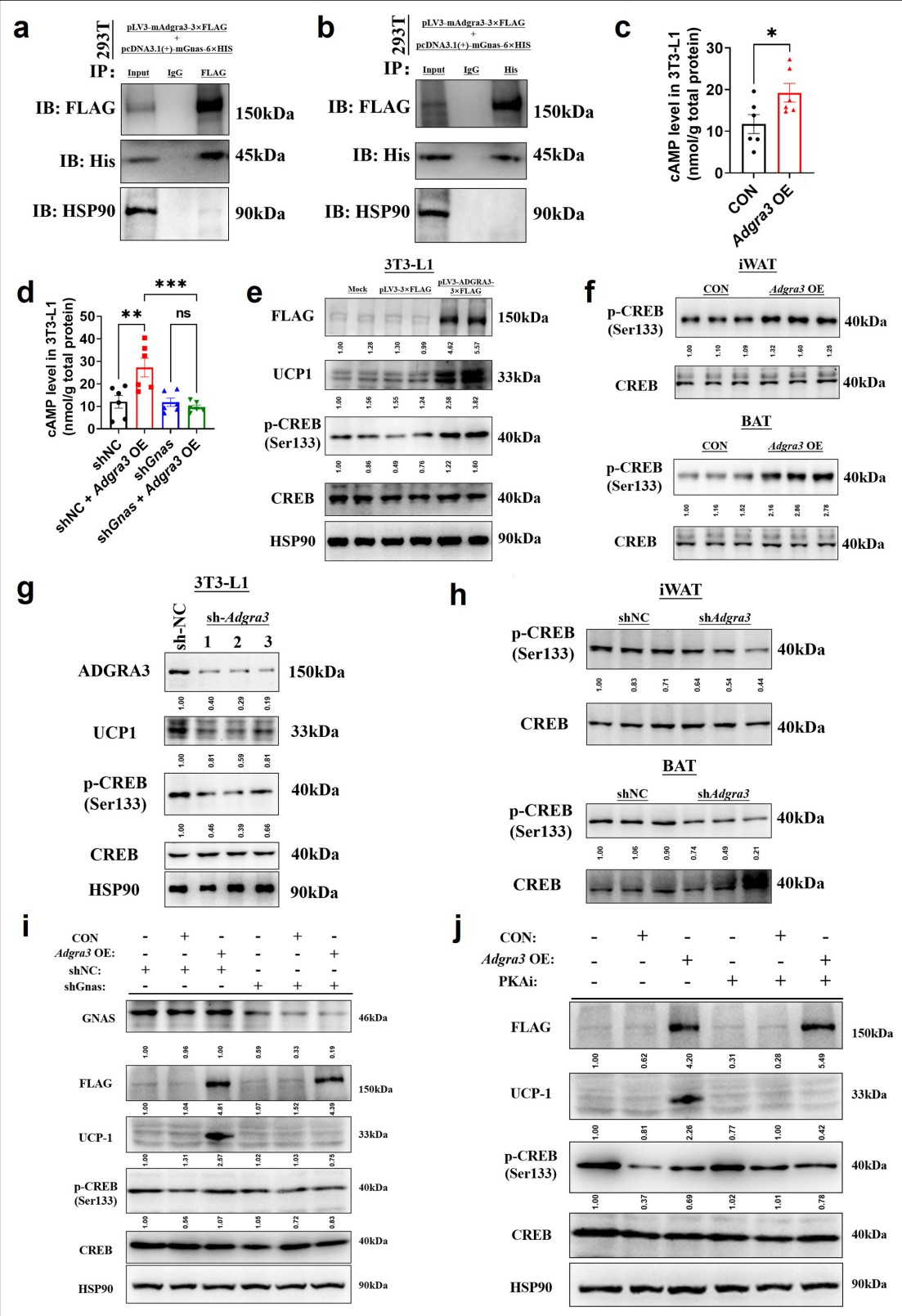

**Figure 5.** ADGRA3 promotes the biogenesis of beige adipocytes via the Gs-PKA-CREB axis. (**A–B**) Western-blot analysis for level of ADGRA3–3×FLAG, GNAS-6 ×HIS and HSP90 proteins in 293T transfected with different plasmids. (**C–D**) The level of cAMP in 3T3-L1. An ELISA kit was used to measure the level of cAMP (N=6 for each group). (**E, G and I–J**) Western-blot analysis for level of ADGRA3, ADGRA3–3×FLAG, UCP1, p-CREB and CREB protein in 3T3-L1 mature beige-like adipocytes. (**F and H**) Western-blot analysis for level of p-CREB and CREB proteins in iWAT and BAT from differently treated

*Figure 5 continued on next page*

*Figure 5 continued*

mice. PKAi, protein kinase A inhibitor, 20 μM H-89. All data are presented as mean ± *SEM*. Statistical significance was determined by unpaired two-tailed student's t-test (**C**) and one-way ANOVA (**D**).

The online version of this article includes the following source data and figure supplement(s) for figure 5:

**Source data 1.** Raw uncropped blots for *Figure 5*.

**Source data 2.** Uncropped and labeled blots for *Figure 5*.

**Source data 3.** Numerical source data for *Figure 5*.

**Figure supplement 1.** ADGRA3 was not observed to bind to G$_i$, G$_q$ and G$_{12}$.

**Figure supplement 1—source data 1.** Raw uncropped blots for *Figure 5—figure supplement 1*.

**Figure supplement 1—source data 2.** Uncropped and labeled blots for *Figure 5—figure supplement 1*.

**Figure supplement 1—source data 3.** Numerical source data for *Figure 5—figure supplement 1*.

hesperetin is dependent on PKA. These findings suggest that hesperetin exerts an induction effect on biogenesis of beige adipocytes via ADGRA3-G$_s$-PKA-CREB axis.

## Hesperetin: a potential ADGRA3 agonist that activates the adipose thermogenic program and counteracts metabolic disease dependent on ADGRA3

To identify whether hesperetin induces adipose thermogenesis and improves the metabolic homeostasis against obesity via ADGRA3, shNC mice or sh*Adgra3* mice were treated with hesperetin and fed with a HFD (*Figure 7A*). Hesperetin was found to alleviate the growth of body weight (*Figure 7B*) during the HFD feeding and the weight of iWAT, eWAT, BAT and liver weight ratio (*Figure 7C*), which was dependent on ADGRA3. It is noteworthy that the food consumption of sh*Adgra3* mice slightly surpassed that of shNC mice, while the administration of hesperetin remained uninfluential on their dietary intake (*Figure 7—figure supplement 1A*). Hesperetin increased body temperature (*Figure 7D*) and BAT temperature (*Figure 7E*) in shNC mice, which were significantly blunted in sh*Adgra3* mice. The levels of serum fT4 were measured to evaluate the consequences of hesperetin treatment on thyroid activity, which indicated that hesperetin treatment does not activate thyroid activity (*Figure 7—figure supplement 1B*).

Concurrently, hesperetin induced a decline in TG level in both serum (*Figure 7—figure supplement 1C*) and liver (*Figure 7—figure supplement 1D*), and also reduced the area of adipocytes in iWAT (*Figure 7—figure supplement 1E*) and BAT (*Figure 7—figure supplement 1F*). However, these effects were absent in sh*Adgra3* mice. Moreover, the expression level of UCP1 were elevated in both iWAT (*Figure 7F and H*) and BAT (*Figure 7G and I*) after hesperetin treatment in shNC mice but not in sh*Adgra3* mice. The results indicated that hesperetin treatment resulted in a substantial decrease in lipid droplet size and a significant increase in mitochondria quantity in BAT (*Figure 7J*). However, this browning effect was weakened in sh*Adgra3* mice. These findings suggest that hesperetin is sufficient to orchestrate the hallmarks of thermogenesis in mice, which is dependent on ADGRA3.

We then investigated the metabolic impact of hesperetin treatment. The GTT presented that hesperetin improved the glucose resistance of HFD mice which showed no effect in sh*Adgra3* mice (*Figure 7K*). The ITT showed that hesperetin alleviated the insulin resistance of HFD mice which showed no significance in sh*Adgra3* mice (*Figure 7L*). Moreover, the fasting serum insulin level was reduced after hesperetin treatment (*Figure 7M*) and the HOMA-IR also showed a moderate improvement (*Figure 7N*), which were dependent on ADGRA3. Taken together, hesperetin activates the adipose thermogenic program and improves the metabolic homeostasis in diet-induced obese mice against obesity and insulin resistance in vivo, which is ADGRA3 dependent.

## ADGRA3 overexpression induces the biogenesis of human beige adipocytes in vitro

Given the elevated expression level of *ADGRA3* compared to *ADRB3* in human adipose tissue (*Figure 1E*, *Figure 1—figure supplement 1E*), we induced human adipose-derived mesenchymal stem cells (hADSCs) and mouse adipose-derived stromal vascular fraction (SVF) into adipocytes to evaluate the effect of ADGRA3 on human adipocytes. The results showed that *ADGRA3* knockdown

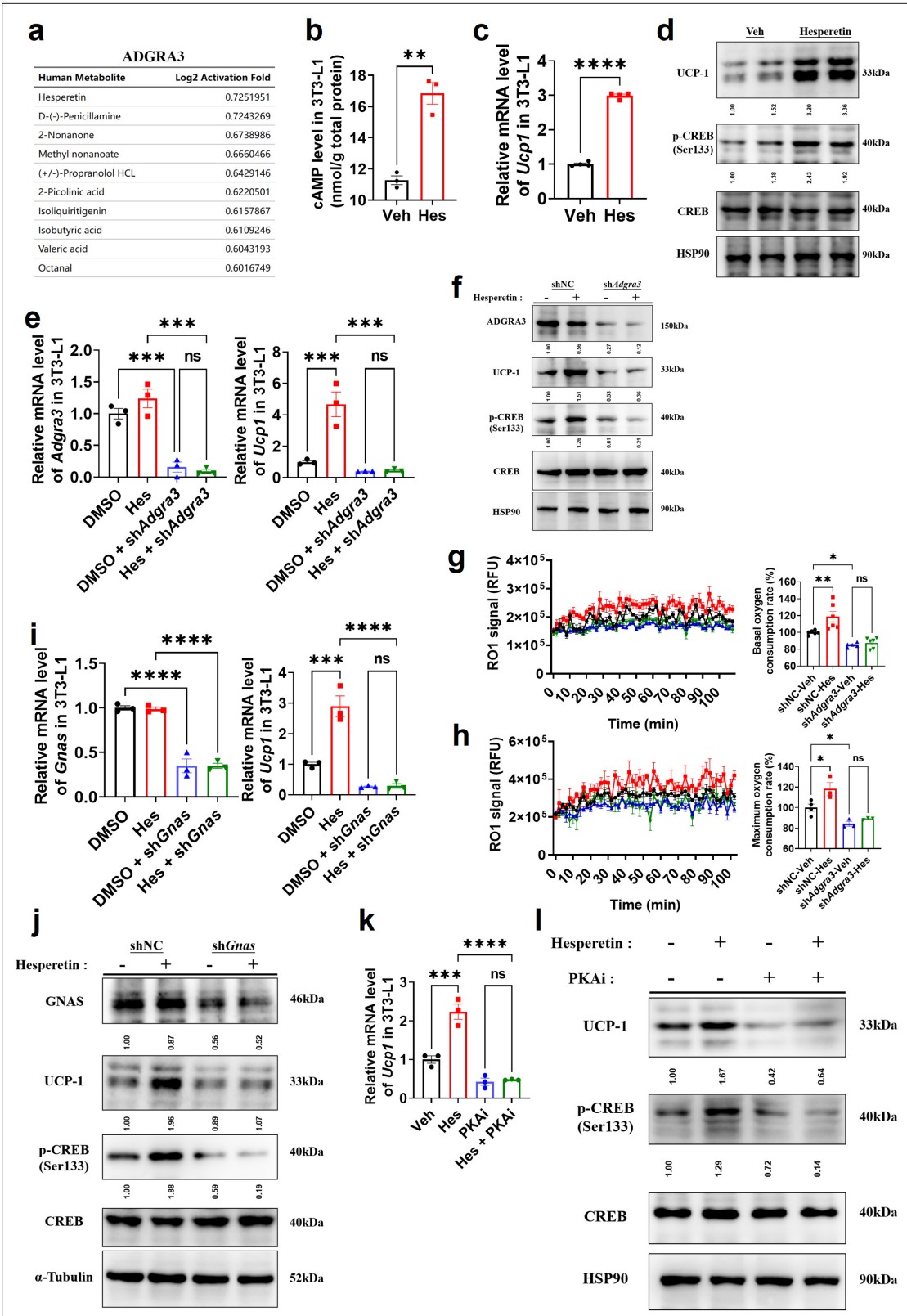

**Figure 6.** Hesperetin promotes the biogenesis of beige adipocytes via ADGRA3-G$_s$-PKA-CREB axis. (**A**) Table of human metabolites with the ability to activate ADGRA3, from the PRESTO-Salsa database. (**B**) The level of cAMP in 3T3-L1. ELISA kit was used to measure the level of cAMP (N=3 for each group). (**C, E, I and K**) qPCR analysis of *Adgra3*, *Gnas* and *Ucp1* in 3T3-L1 mature beige-like adipocytes (N=3 for each group). (**D, F, J and L**) Western-blot analysis for level of ADGRA3, GNAS, UCP1, p-CREB and CREB protein in 3T3-L1 mature beige-like adipocytes. (**G**) When 3T3-L1 mature beige-like

*Figure 6 continued on next page*

*Figure 6 continued*

adipocytes were treated with shNC, sh*Adgra3*, or Hesperetin, fluorescence of the oxygen probe (RO1) in the cells was monitored and the rate of basal oxygen consumption was analyzed (N=5 for sh*Adgra3*-Veh; N=6 for each other group). (**H**) When FCCP-treaded 3T3-L1 mature beige-like adipocytes were treated with shNC, sh*Adgra3*, or Hesperetin, fluorescence of the oxygen probe (RO1) in the cells was monitored and the rate of maximum oxygen consumption was analyzed (N=4 for shNC-Veh; N=3 for each other group). Hes, 10 μM Hesperetin; PKAi, protein kinase A inhibitor, 20 μM H-89. All data are presented as mean ± *SEM*. Statistical significance was determined by unpaired two-tailed student's t-test (**B–C**) and one-way ANOVA (**E, G–I and K**).

The online version of this article includes the following source data for figure 6:

**Source data 1.** Raw uncropped blots for *Figure 6*.

**Source data 2.** Uncropped and labeled blots for *Figure 6*.

**Source data 3.** Numerical source data for *Figure 6*.

led to a diminished expression of *UCP1* (*Figure 8A and E*), whereas its overexpression elicited an enhancement in *UCP1* expression (*Figure 8B and F*). Furthermore, Mito-Tracker and lipid droplet fluorescence staining illuminated a notable increase in lipid droplet count accompanied by a decrease in mitochondrial number following *ADGRA3* knockdown (*Figure 8C*). Conversely, *ADGRA3* overexpression resulted in a visible surge in mitochondrial quantity and a marked reduction in lipid droplet presence (*Figure 8D*). To further verify whether hesperetin induces the expression of UCP1 via ADGRA3, we treated mouse primary adipocytes with hesperetin and sh*Adgra3*, respectively. We found that the induction effect of hesperetin on UCP1 is eliminated when *Adgra3* is knocked down (*Figure 8G–H*) in primary cultures.

## Discussion

In the present study, we have elucidated a novel role of ADGRA3 and hesperetin in inducing the development of beige adipocytes through the activation of the $G_s$-PKA-CREB signaling pathway. ADGRA3 is responsible for the activation of the adipose thermogenic program and plays a significant role in maintaining systemic glucose homeostasis. Additionally, the development of beige adipocytes induced by hesperetin is contingent upon the presence of ADGRA3. The novelty of this study is the discovery that ADGRA3 plays a role in the advancement of beige fat and the regulation of metabolic homeostasis. This suggests that targeting the ADGRA3-$G_s$-PKA-CREB signaling pathway could potentially be a therapeutic approach for obesity and related metabolic disorders.

The induction of beige fat has been investigated as a potentially effective therapeutic approach in combating obesity (*Harms and Seale, 2013*). A clinical trial revealed that treatment with the chronic β3-AR agonist mirabegron leads to an increase in human brown fat, HDL cholesterol, and insulin sensitivity (*O'Mara et al., 2020*). Subsequently, Blondin et al discovered that oral administration of mirabegron only elicits an increase in BAT thermogenesis when administered at the maximal allowable dose, indicating that human brown adipocyte thermogenesis is primarily driven by β2-adrenoceptor (β2-AR) stimulation (*Blondin et al., 2020*). Consistent with this finding, we found much higher levels of ADRB2 expression in human white adipose tissue than ADRB3 (*Figure 1—figure supplement 1E*). Furthermore, a recent study has demonstrated that simultaneous activation of β2-AR and β3-AR enhances whole-body metabolism through beneficial effects on skeletal muscle and BAT (*Talamonti et al., 2024*).

While the promotion of thermogenesis in brown and beige adipocytes in rodents was effectively achieved by the β3-adrenoceptor agonist, the clinical implications of this finding appear to be unfeasible in humans due to the low efficacy of β3-adrenoceptor agonists (*Fisher et al., 2012*; *Kajimura et al., 2015*). It is of utmost importance to investigate alternative therapeutic targets that can effectively and selectively enhance beige adipogenesis in order to combat obesity and its related metabolic disorders. In this study, we have identified ADGRA3 as a novel GPCR therapeutic target that exhibits high expression in human adipocytes. In human adipose tissue, ADGRA3 is expressed at a lower level than ADRB2 (*Figure 1—figure supplement 1* E), which has been shown to be the main receptor mediating adrenergic activation of thermogenesis in human brown adipocytes (*Blondin et al., 2020*). Nevertheless, given ADRB2's pivotal role in bronchodilation and vasodilation (*Hizawa, 2011*; *Morell et al., 2021*), we believe that ADGRA3 has the potential to be an alternative target for inducing adipose thermogenesis. Overall, these findings suggest that ADGRA3, when overexpressed or stimulated by its potential agonist, hesperetin, can induce the biogenesis of beige fat.

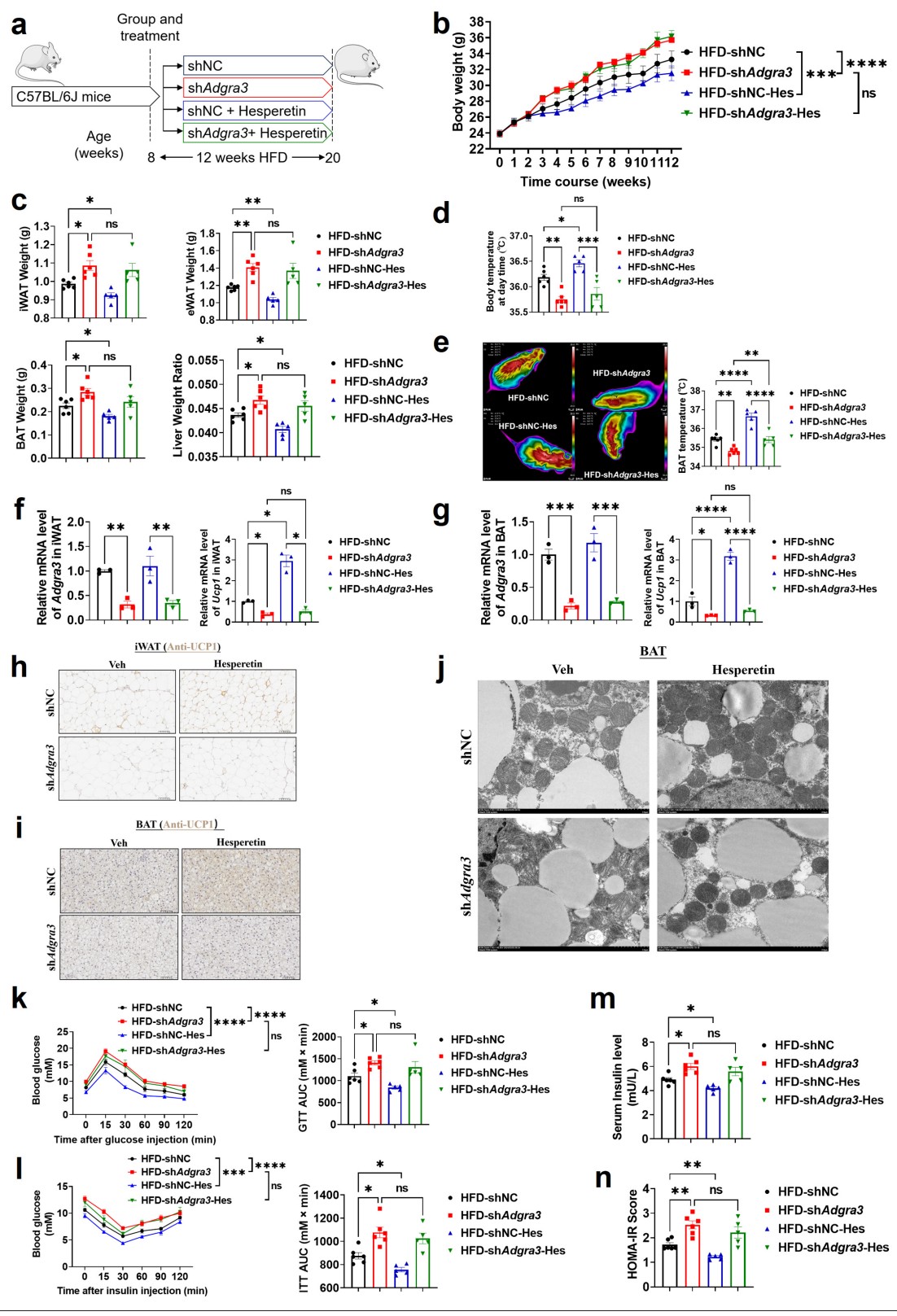

**Figure 7.** Hesperetin activated the adipose thermogenic program and facilitated metabolic homeostasis in mice with diet-induced obesity (DIO) dependent on ADGRA3. (**A**) Experimental schematic. Different treated C57BL/6 J mice were fed with a HFD for 12 weeks. (**B–D**) Changes in body mass (**B**), tissue weight (**C**) and body temperature (**D**) of different treated mice (N=6 for HFD-shNC and HFD-sh*Adgra3*; N=5 for HFD-shNC-Hes and HFD-sh*Adgra3*-Hes). (**E**) Thermal image and BAT temperature of different treated mice (N=6 for HFD-shNC and HFD-sh*Adgra3*; N=5 for HFD-shNC-

*Figure 7 continued on next page*

*Figure 7 continued*

Hes and HFD-sh*Adgra3*-Hes). (**F–G**) qPCR analysis of *Adgra3* and *Ucp1* in iWAT (**F**) and BAT (**G**) from different treated mice (N=3 for each group). (**H–I**) Representative images of iWAT (H; Scale bars, 50 µm.) and BAT (I; Scale bars, 50 µm.) stained with UCP1. (**J**) Transmission electron microscope photograph of BAT from different treated mice (Scale bars, 2 µm.). (**K**) Glucose tolerance test (GTT) was conducted by intraperitoneal injection of glucose (1 g/kg) and measurement of blood glucose concentration with a OneTouch Ultra Glucometer at designed time points in six hours fasted mice (N=6 for HFD-shNC and HFD-sh*Adgra3*; N=5 for HFD-shNC-Hes and HFD-sh*Adgra3*-Hes). (**L**) Insulin tolerance test (ITT) was done by intraperitoneal injection of insulin (1 U/kg) and measurement of blood glucose concentration by a OneTouch Ultra Glucometer at designed time points in six hours fasted mice (N=6 for HFD-shNC and HFD-sh*Adgra3*; N=5 for HFD-shNC-Hes and HFD-sh*Adgra3*-Hes). (**M–N**) The fasting serum insulin (**M**) and HOMA-IR (**N**) in different treated mice (N=6 for HFD-shNC and HFD-sh*Adgra3*; N=5 for HFD-shNC-Hes and HFD-sh*Adgra3*-Hes). HOMA-IR=Fasting glucose level (mmol/L) * Fasting insulin level (mIU/L) /22.5. HFD, high-fat diet; iWAT, inguinal white adipose tissue; BAT, brown adipose tissue; GTT, Glucose tolerance test; ITT, Insulin tolerance test; HOMA-IR, homeostasis model assessment of insulin resistance; Hes, Hesperetin. All data are presented as mean ± *SEM*. Statistical significance was determined by one-way ANOVA (**C–G and M–N**) and two-way ANOVA (**B and K–L**).

The online version of this article includes the following source data and figure supplement(s) for figure 7:

**Source data 1.** Numerical source data for *Figure 7*.

**Figure supplement 1.** Characterization of wild-type and *Adgra3*-knockdown mice after hesperetin treatment.

**Figure supplement 1—source data 1.** Numerical source data for *Figure 7—figure supplement 1*.

Hesperetin has been reported to attenuate the age-related metabolic decline, reduce fat and improve glucose homeostasis in naturally aged mice (*Yeh et al., 2022*). Previous studies showed that hesperetin improved glycemic control (*Yeh et al., 2022*; *Xue et al., 2016*) and was involved in adipocyte differentiation (*Subash-Babu and Alshatwi, 2015*), but whether hesperetin induces the biogenesis of beige adipocyte was uncertain. Previously, the influence of hesperetin on ADGRA3 has remained unreported. In this study, we screened hesperetin as a potential agonist for ADGRA3 by using the PRESTO-Salsa tool as well as discovered that hesperetin has an agonist effect on ADGRA3 through a series of experiments. This study focuses on the regulatory effect of hesperetin on adipose thermogenesis and explores whether this effect is dependent upon ADGRA3. As such, we refrained from conducting further investigations into other potential effects of hesperidin, including its potential role in antioxidant and in apoptosis.

In previous reports, male mice deficient in ADGRA3 showed obstructive azoospermia with high penetrance (*Nybo et al., 2023*). Moreover, a GWAS identified an SNP located in the downstream region of *ADGRA3* as a genomic locus associated with body weight in chickens, suggesting that the ADGRA3 is a potential regulator of body weight (*Cha et al., 2021*). Nevertheless, the agonist and the downstream signal axis of ADGRA3 remain unclear as well as the effects of ADGRA3 on adipose thermogenesis and glucose homeostasis have not been explored. Consequently, our study has confirmed that the knockdown of *Adgra3* exacerbates obesity and disrupts glucose homeostasis. Additionally, both the overexpression of *Adgra3* and the administration of hesperetin have been found to stimulate the biogenesis of beige adipocytes through the ADGRA3-$G_s$-PKA-CREB signaling pathway and improve glucose homeostasis.

Given the consideration that the non-targeted nanoparticle approach utilized in this study for modulating *Adgra3* expression levels in vivo alter *Adgra3* expression in tissues beyond adipose tissue (*Figure 3—figure supplement 1A-B*, *Figure 4—figure supplement 1B-C*), notably the liver and skeletal muscle, the construction of *Adgra3* adipose tissue-specific knockout/overexpression mouse models is imperative for a more nuanced understanding of the precise mechanisms underlying the influence of on adipose thermogenesis. Furthermore, it is crucial to highlight that the observed decrease in TG levels in both serum and liver (*Figure 4—figure supplement 2C–D*) might be attributed to the significant increase in *Adgra3* expression in the liver, which is a consequence of the nanoparticle-mediated overexpression of *Adgra3*. While the exact mechanism remains to be fully elucidated, this correlation suggests a potential link between *Adgra3* overexpression in the liver and reduced TG levels in the serum. We will employ more sophisticated models in subsequent studies to further elucidate the effects of ADGRA3 on adipose thermogenesis and metabolic homeostasis. Nevertheless, our findings underlie a potential therapeutic feature of ADGRA3 and hesperetin in obesity and the associated metabolic diseases from the thermogenic viewpoint of beige fat.

In conclusion, the activation of the $G_s$-PKA-CREB axis by ADGRA3 has been found to induce adipose thermogenesis, promote lipid metabolism, and alleviate lipid accumulation in adipose tissues (*Figure 9*). Furthermore, the induction of beige adipocyte biogenesis by hesperetin occurs through

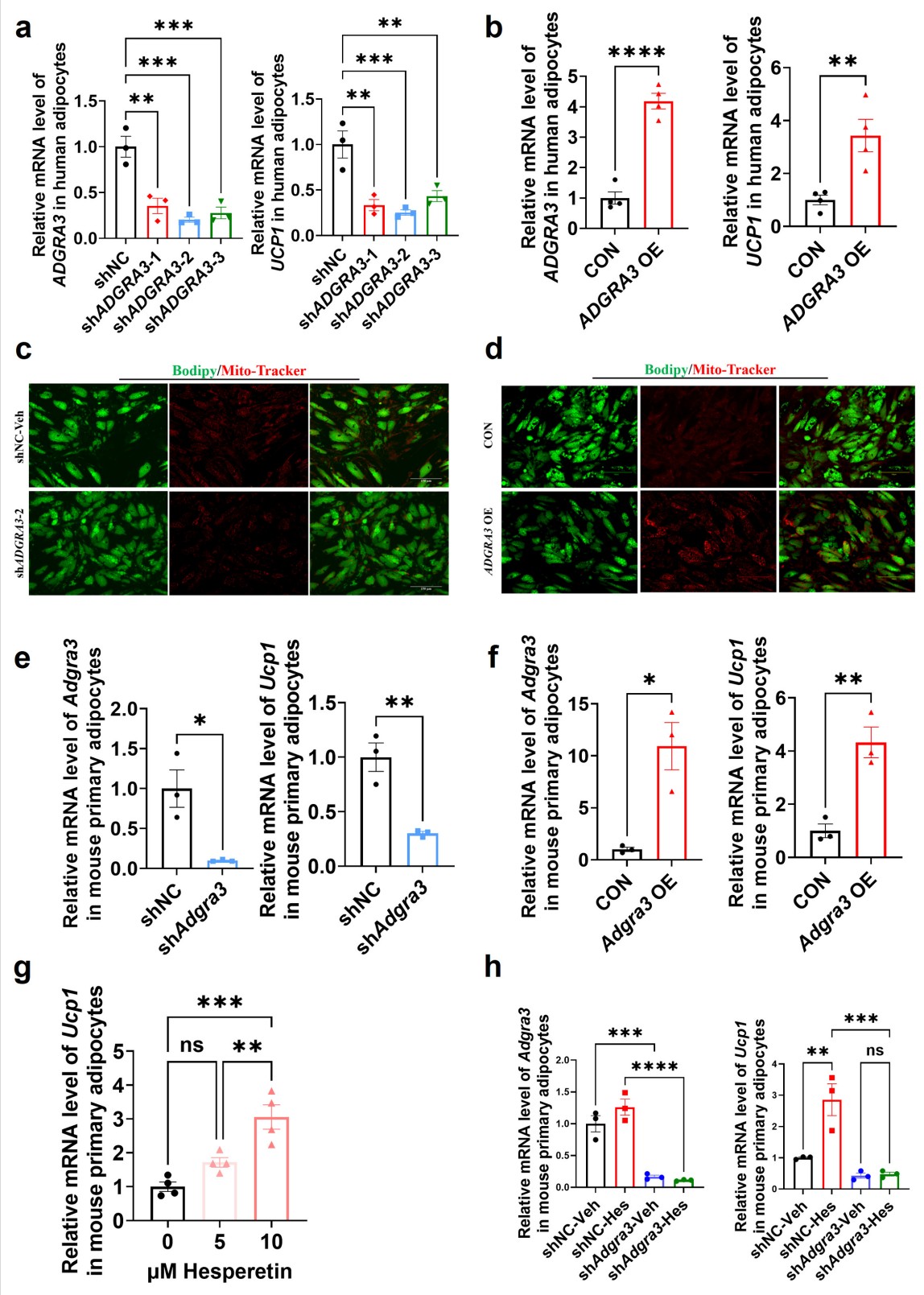

**Figure 8.** ADGRA3 overexpression induces the beiging of human adipocytes. (**A–B**) qPCR analysis of *ADGRA3* and *UCP1* genes in adipocytes induced from human adipose-derived mesenchymal stem cells (hADSCs) (A: N=3 for each group; B: N=4 for each group). (**C–D**) Bodipy green staining for lipid droplet and Mito-Tracker red staining for mitochondria in adipocytes induced from hADSCs. Scale bars, 150 μm. (**E–H**) qPCR analysis of *Adgra3* and *Ucp1* genes in mouse primary adipocytes induced from stromal vascular fraction (SVF) of WAT (E-H: N=3 for each group). sh*ADGRA3* (pLKO.1-U6-

*Figure 8 continued*

sh*ADGRA3*-(1/2/3) plasmid encapsulated in nanomaterials), shNC (pLKO.1-U6-shNC plasmid encapsulated in nanomaterials), *ADGRA3* OE (pLV3-CMV-*ADGRA3*(human)–3×FLAG plasmid encapsulated in nanomaterials) or CON (pLV3-CMV-MCS-3×FLAG plasmid encapsulated in nanomaterials). All data are presented as mean ± *SEM*. Statistical significance was determined by unpaired two-tailed student's t-test (**B and E–F**) and one-way ANOVA (**A and G–H**).

The online version of this article includes the following source data for figure 8:

**Source data 1.** Numerical source data for *Figure 8*.

the ADGRA3-G$_s$-PKA-CREB axis. Given the importance of identifying signaling pathways that induce beige fat and alleviate obesity-related dysfunction in adipose tissue, our research findings suggest that hesperetin and activation of the intracellular signaling of ADGRA3 could serve as a promising and innovative therapeutic approach.

# Materials and methods

## Key resources table

| Reagent type (species) or resource | Designation | Source or reference | Identifiers | Additional information |
| --- | --- | --- | --- | --- |
| Cell line (*Homo-sapiens*) | 293T | Cell Bank of the Chinese Academy of Sciences in Shanghai | SCSP-502 | |
| Cell line (*M. musculus*) | 3T3-L1 | Cell Bank of the Chinese Academy of Sciences in Shanghai | SCSP-5038 | |

*Continued on next page*

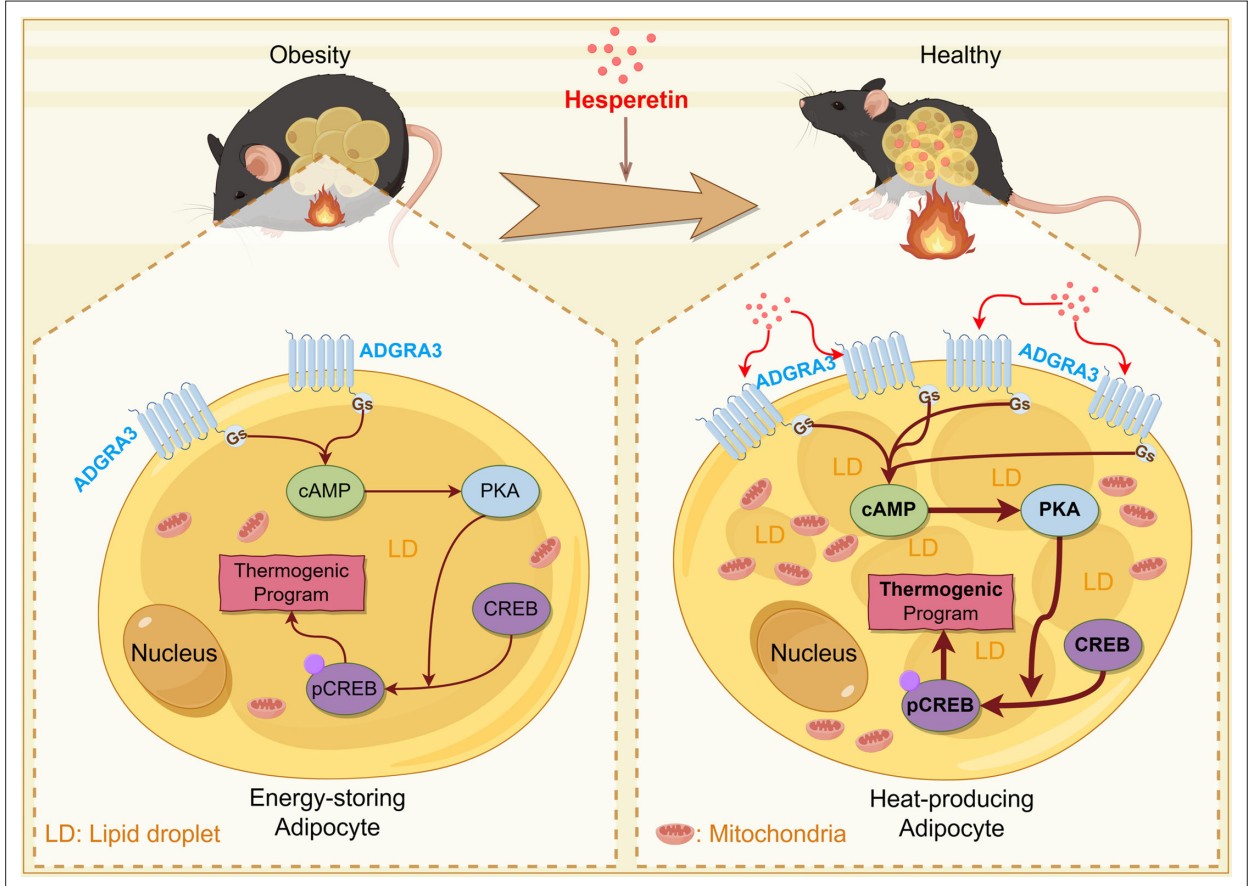

**Figure 9.** Graphical abstract. This schema summarizes the main roles and functions of ADGRA3 in lipid metabolism. ADGRA3 promotes the biogenesis of beige adipocytes via the Gs-PKA-CREB axis. This figure was drawn by Figdraw (Copyright ID: IWAIA0d9f9).

*Continued*

| Reagent type (species) or resource | Designation | Source or reference | Identifiers | Additional information |
|---|---|---|---|---|
| Antibody | anti-HSP90 (Rabbit monoclonal) | Cell Signaling Technology | Cell Signaling Technology Cat# 4877, RRID:AB_2233307 | WB (1:1000) |
| Antibody | anti-α-tubulin (Mouse monoclonal) | Proteintech | Proteintech Cat# 66031-1-Ig, RRID:AB_11042766 | WB (1:10000) |
| Antibody | anti-ADGRA3 (Rabbit polyclonal) | Proteintech | Proteintech Cat# 11912-1-AP, RRID:AB_2877804 | WB (1:1000) |
| Antibody | anti-UCP1 (Rabbit polyclonal) | Abcam | Abcam Cat# ab10983, RRID:AB_2241462 | WB (1:1000 for iWAT and cells or 1:10000 for BAT) |
| Antibody | anti-FLAG-tag (Mouse monoclonal) | Beyotime | Beyotime Cat# AF2855, RRID:AB_3674126 | WB (1:2000) |
| Antibody | anti-HIS-tag (Mouse monoclonal) | Beyotime | Beyotime Cat# AF2879, RRID:AB_3674127 | WB (1:2000) |
| Antibody | anti-p-CREB (Rabbit polyclonal) | Beyotime | Beyotime Cat# AF5785, RRID:AB_3674128 | WB (1:1000) |
| Antibody | anti-CREB (Rabbit polyclonal) | Beyotime | Beyotime Cat# AF6566, RRID:AB_3674129 | WB (1:1000) |
| Antibody | anti-FLAG-tag (Mouse monoclonal) | Servicebio | ServiceBio Cat# GB15938-100, RRID:AB_3674125 | IP (1:50) |
| Antibody | anti-HIS-tag (Mouse monoclonal) | Servicebio | ServiceBio Cat# GB151251, RRID:AB_3665294 | IP (1:50) |
| Biological sample (*Homo-sapiens*) | hADSCs | National Stem Cell Translational Resource Center | ZB10DGAC | |
| Biological sample (*M. musculus*) | SVF | Center of Laboratory Animal at Sun Yat-sen University | | Freshly isolated from C57BL/6 J (Male) |
| Other | Lipo8000 | Beyotime | C0533 | The nanomaterials mentioned in this article refer to Lipo8000 reagent, a highly efficient transfection reagent based on nanomaterials |

## Nanomaterials

The nanomaterials mentioned in this article refer to Lipo8000 reagent (Beyotime, C0533), a highly efficient transfection reagent based on nanomaterials, unless otherwise specified.

## Mice

Wild-type (WT) C57BL/6 J mice were obtained from the Center of Laboratory Animal at Sun Yat-sen University. All mice were housed in the Sun Yat-sen University Laboratory Animal Center, where they were subjected to a 12 hr light-dark cycle and maintained at a controlled environmental temperature of 21±1°C. Eight-week-old male C57BL/6 J mice were fed with a normal chow diet (NCD) or a high fat diet (HFD, 60% kcal) for 12 weeks to render mice obese. With the exception of mice fed with a HFD, male mice at the age of eight weeks were utilized in all experimental procedures.

For the knockdown and over-expression experiments of *Adgra3* in mice fed with a NCD, the following procedures were conducted: sh*Adgra3* (pLKO.1-U6-sh*Adgra3*-2 plasmid encapsulated in nanomaterials) and shNC (pLKO.1-U6-shNC plasmid encapsulated in nanomaterials) were injected intraperitoneally (i.p.) for knockdown experiments, while *Adgra3* OE (pLV3-CMV-*Adgra3* (mouse)–3×FLAG plasmid encapsulated in nanomaterials) and CON (pLV3-CMV-MCS-3×FLAG plasmid encapsulated in nanomaterials) were injected i.p. for over-expression experiments. The frequency of the sessions was twice a week over a period of 4 weeks. For the knockdown and over-expression experiments of *Adgra3* in mice fed with a HFD, the following procedures were conducted: sh*Adgra3* and shNC were injected intraperitoneally (i.p.) for knockdown experiments, while *Adgra3* OE and CON

were injected i.p. for over-expression experiments. The frequency of the sessions was twice a week over a period of 12 weeks. For the local over-expression or knockdown of *Adgra3* in BAT of mice fed with a NCD, *Adgra3* OE, sh*Adgra3*, CON and shNC were injected locally into BAT once.

For the treatment with a selective β3-adrenoceptor agonist, CL-316,243 (hereafter referred to as CL), mice fed a HFD were injected intraperitoneally (i.p.) with CL (1 mg/kg daily) for 7 days. For the treatment with hesperetin (Hes), hesperetin is dissolved in drinking water (200 mg/L) and water were available ad libitum.

Intraperitoneal injections of glucose (2 g/kg for mice fed with a NCD and 1 g/kg for mice fed with a HFD) or insulin (0.5 U/kg for mice fed with a NCD and 1 U/kg for mice fed with a HFD) were administered. At the designated time points of 0 min, intraperitoneal glucose or insulin tolerance tests were conducted on mice that had been fasted for six hours. After administration, the blood glucose concentration was assessed at specific time intervals using a OneTouch Ultra Glucometer. Finally, the animals were euthanized, followed by the collection of tissue samples. Cohorts of ≥3 mice per genotype or treatment were assembled for all in vivo studies. All in vivo studies were repeated two to three independent times. All procedures related to animal feeding, treatment and welfare were conducted at Sun Yat-sen University Laboratory Animal Center. All the animal experiments were conducted with the approval of the Animal Care and Use Committee of Sun Yat-sen University (Approval ID: SYSU-IACUC-MED-2023-B082). This study was conducted in accordance with the ethical principles derived from the Declaration of Helsinki and Belmont Report and was approved by the review board of Sun Yat-sen University (Guangzhou, China).

## Stromal vascular fraction (SVF) and mature adipocytes isolation

SVF from inguinal white adipose tissue (iWAT) and BAT of WT male mice at 4 weeks of age were washed with PBS, minced and digested with 0.1% type II collagenase in Dulbecco's modified eagle medium (DMEM) containing 3% BSA and 25 µg/ml DNase Ⅰ for 30 min at 37°C. During the digestion, the mixed solution was shaken by a hand every 5 min. The mixed solution was filtered through a 70 µm cell strainer and then centrifuged at 500 × *g* for 5 min at 4°C. The floating mature adipocytes were collected for subsequent analysis, and the pellets containing SVF were resuspended in red blood cell lysis buffer for 5 min at 37°C. Cells were centrifuged at 500 × *g* for 10 min at 4°C and the SVF pellets were collected for subsequent analysis.

## Cell culture

3T3-L1 and 293T cell lines were purchased from the Cell Bank of the Chinese Academy of Sciences in Shanghai, and both were identified by STR and tested negative for mycoplasma. Human adipose-derived mesenchymal stem cells (hADSCs) were purchased from the National Stem Cell Translational Resource Center. 3T3-L1 cells were culture and grown to confluence in high-glucose DMEM supplemented with 10% newborn calf serum (NCS). 293T cells were culture and grown to confluence in high-glucose DMEM supplemented with 10% FBS. Confluent 3T3-L1 and SVF pre-adipocytes were induced into mature beige-like adipocytes with 0.5 mM isobutyl methylxanthine (IBMX), 1 µM dexamethasone, 5 µg/ml insulin, 1 nM 3, 3', 5-Triiodo-L-thyronine (T3), 125 µM indomethacin and 1 µM rosiglitazone in high-glucose DMEM containing 10% FBS for 2 days, then treated with high-glucose DMEM containing 5 µg/ml insulin, 1 nM T3, 1 µM rosiglitazone and 10% FBS for 6 days and cultured with high-glucose DMEM containing 10% FBS for 2 days. hADSCs were seeded on plates coated with 0.1% gelatin and culture and grown to confluence in human mesenchymal stem cells (hMSCs) specialized culture medium (ZQ-1320). Confluent hADSCs were induced into mature human adipocytes with adipogenic induction medium (PCM-I-004) according to the manufacturer's instructions.

The sh*Adgra3*, sh*Gnas* (pLKO.1-U6-sh*Gnas* plasmid encapsulated in nanomaterials) and shNC were added to 3T3-L1 mature beige-like adipocytes for 72 hr. The sh*ADGRA3* (pLKO.1-U6-sh*ADGRA3*-(1/2/3) plasmid encapsulated in nanomaterials) and shNC were added to human adipocytes for 72 hr. The pcDNA3.1(+)-m*Gnas*-6×His (mixture of pcDNA3.1(+)-*Gnas*(mouse)–6×His plasmid and transfection reagent), pcDNA3.1(+)-m*Gnai1*–6×His (mixture of pcDNA3.1(+)-*Gnai1*(mouse)–6×His plasmid and transfection reagent), pcDNA3.1(+)-m*Gnaq*-6×His (mixture of pcDNA3.1(+)-*Gnaq*(mouse)–6×His plasmid and transfection reagent), pcDNA3.1(+)-m*Gna12*–6×His (mixture of pcDNA3.1(+)-*Gna12*(mouse)–6×His plasmid and transfection reagent), *Adgra3* OE and CON were added to 3T3-L1 mature beige-like adipocytes or 293T for 48 hr. The *ADGRA3* OE (pLV3-CMV-*ADGRA3*(human)–3×FLAG

plasmid encapsulated in nanomaterials) and CON were added to human adipocytes for 48 hr. Hesperetin (10 μM) and PKAi (protein kinase A inhibitor, 20 μM H-89) was added to 3T3-L1 mature beige-like adipocytes for 48 hr. All in vitro studies were repeated two to three independent times.

## Construction of plasmid

The pLV3-CMV-*Adgra3*(mouse)–3×FLAG, pLV3-CMV-*ADGRA3*(human)–3×FLAG, pLKO.1-U6-sh*ADGRA3*-(1/2/3), pLV3-CMV-MCS-3×FLAG, pcDNA3.1(+)-*Gnas*(mouse)–6×His, pcDNA3.1(+)-*Gnai1*(mouse)–6×His, pcDNA3.1(+)-*Gnaq*(mouse)–6×His and pcDNA3.1(+)-*Gna12*(-mouse)–6×His plasmids were purchased from Shenzhen Yanming Biotechnology Co., LTD. The pLKO.1-U6-sh*Adgra3*-(1/2/3) and pLKO.1-U6-shNC plasmids were purchased from Guangzhou Hanyi Biotechnology Co., LTD.

## Temperature measurements

The body temperature was measured at 9:00 using a rectal probe connected to a digital thermometer.

## Real-time polymerase chain reaction (PCR)

Total RNA from tissue or cells was extracted with Trizol reagent. RNA concentration was measured by a NanoDrop spectrometer. 1000 ng total RNA was reverse transcribed into cDNA by All-in-One RT SuperMix (G3337). Real-time PCR analysis using SYBR-Green fluorescent dye was performed with Step One Plus RT PCR System. Primers used for real-time PCR were listed in *Supplementary file 1*.

## Histology and immunohistochemistry

Subcutaneous, epididymal white adipose tissue, interscapular brown adipose tissue and liver were fixed in 4% paraformaldehyde. Tissues were embedded with paraffin and sectioned by microtome. The slides were stained with hematoxylin and eosin (HE) using a standard protocol. For UCP1 and ADGRA3 immunohistochemistry, slides of various tissue were blocked with goat serum for 1 hr. Subsequently, the slides were incubated with anti-UCP1 (1:1000; ab10983) or anti-ADGRA3 (1:200; 11912–1-AP) overnight at 4°C followed by detection with the EnVision Detection Systems. Hematoxylin was used as counterstain.

## Western-blot

Tissues and cells were lysed in RIPA buffer supplemented with 1 mM PMSF and protease inhibitor cocktail (K1007). The protein concentration was measured by the BCA protein assay kit (BL521), and total cellular protein (25 μg) was subject to Western-blot analysis. The protein transferred to the PVDF membrane was probed with primary antibodies specific for HSP90, α-tubulin, ADGRA3, UCP1, FLAG-tag, HIS-tag, p-CREB and CREB overnight at 4 °C. Except FLAG-tag protein and HIS-tag protein, after being incubated with HRP conjugated secondary antibody, proteins were detected with chemiluminescence using Immobilon Western HRP Substrate on ChemiDoc MP Imaging System. The ImageJ software was used for gray scanning. For all Western-blots, each lane represented an independent sample and all experiments were replicated 2–3 times.

## IP assay

HEK293T cells were transfected using PEI 40 K transfection reagent (G1802) with indicated cDNAs and cultured using the manufacture's protocol. Cells were lysed with IP lysis buffer (G2038) containing protease inhibitor cocktail (K1007). The lysates were precipitated with the FLAG-tag antibody or HIS-tag antibody in the presence of protein A+G agarose (P2055). The precipitants were washed five times with the IP lysis buffer and analyzed by immunoblot with the indicated antibodies.

## Enzyme-linked immunosorbent assay (ELISA)

Mouse cAMP level was detected using a sensitive ELISA kit (MM-0544M2) purchased from Jiangsu Meimian Industrial Co., Ltd. Mouse IP1 level was detected using a sensitive ELISA kit (MM-0790M2) purchased from Jiangsu Meimian Industrial Co., Ltd. Mouse insulin level was detected using a sensitive ELISA kit (MM-0579M1) purchased from Jiangsu Meimian Industrial Co., Ltd. Mouse free tetraiodothyronine (fT4) level was detected using a sensitive ELISA kit (RXJ202449M) purchased from

Quanzhou Ruixin Biological Technology Co., Ltd. All measurements were performed using the manufacture's protocol.

## Bodipy staining

For the lipid staining, the differentiated adipocytes were washed twice with PBS. The cells were then stained with 2 µM BODIPY staining solution (GC42959) for 15 min at 37°C, then washed three times with PBS according to manufacturer's instructions. The stained cells were observed using a fluorescence microscope.

## Mito-Tracker staining

The differentiated adipocytes were incubated with 100 nM Mito-Tracker Red CMXRos (C1049) for 30 min according to manufacturer's instructions. Then cells were washed with PBS and visualized under the confocal microscope.

## Determination of 2-deoxy-D-glucose (2-NBDG) uptake

The differentiated adipocytes were washed twice with PBS. The cells were then incubated with 100 µM 2-NBDG staining solution (HY-116215) for 30 min at 37°C, then washed three times with PBS. The stained cells were observed using a fluorescence microscope.

## Measurement of triacylglycerol (TG)

The triacylglycerol in adipocytes, tissues, and plasma was measured by using Triglyceride Assay Kit (A110-1-1) according to the manufacturer's instructions.

## Transmission electron microscopy

BAT sections were fixed in 2% (vol/vol) glutaraldehyde in 100 mM phosphate buffer, pH 7.2 for 12 hr at 4°C. The sections were then post-fixed in 1% osmium tetroxide, dehydrated in ascending gradations of ethanol and embedded in fresh epoxy resin 618. Ultra-thin sections (60–80 nm) were cut and stained with lead citrate before being examined on the FEI-Tecnai G2 Spirit Twin transmission electron microscope.

## Differential expression analysis

The R package Linear Models for Microarray Data (limma) was used to analyze differential RNA-Sequencing expression. For screening high-expressed G-protein-coupled receptors in mouse BAT, limma was applied in the GSE118849 dataset to screen out BAT-elevated genes. For screening ADGRA3 high-expressed gene sets in human subcutaneous adipose, limma was applied in the human subcutaneous adipose dataset from GTEx Portal to screen out ADGRA3 high-expressed gene sets. Genes highly expressed in human adipocytes were obtained from the human protein atlas database. Genes with the cutoff criteria of $|logFC| \geq 1.0$ and $p < 0.05$ were regarded as differentially expressed genes (DEGs). The DEGs of the GSE118849 dataset and the human subcutaneous adipose dataset were visualized as volcano plots by using the R package ggplot2.

## Functional annotation for genes of interest

To explore DisGeNET, Gene Ontology (GO), WikiPathwas, Kyoto Encyclopedia of Genes and Genomes (KEGG) and Reactome of selected genes, Metascape was used to explore the functions among DEGs, with a cutoff criterion of $p < 0.05$. GO annotation that contains the biological process (BP) subontology, which can identify the biological properties of genes and gene sets for all organisms.

## Gene set enrichment analysis (GSEA)

GSEA was performed to detect a significant difference in the set of genes expressed between the *ADGRA3* high-expressed and *ADGRA3* low-expressed groups in the enrichment of the KEGG collection.

## Oxygen consumption rate (OCR)

The basal oxygen consumption rate of cells was measured using a BBoxiProbe R01 kit (BB-48211) according to the manufacturers' instructions. The maximum oxygen consumption rate of cells was

measured with the addition of FCCP (Trifluoromethoxy carbonylcyanide phenylhydrazone) with a final concentration of 1 μM.

## Infrared thermography

BAT temperature was measured at room temperature by infrared thermography according to previous publications (*Xia et al., 2020*; *Warner et al., 2013*). The same batch of representative infrared images of mice were all captured using a thermal imaging camera (FLIR ONE PRO), measured at the same distance perpendicular to the plane on which the mice were located. To quantify interscapular region temperature, the average surface temperature from a region of the interscapular BAT was taken with FLIR Tools software.

## Statistical analysis

All data are presented as mean ± *SEM*. In this study, outliers that met the three-sigma rule were excluded from analysis, with the exception of those presented in *Figure 1—figure supplement 1E*. Given the possibility that the outliers in *Figure 1—figure supplement 1E* represent extreme expressions of the inherent variability within the population sample, we have chosen to retain these specific outliers for further analysis. Student's t-test was used to compare two groups. One-way analysis of variance (ANOVA) or Two-way ANOVA was applied to compare more than two different groups on GraphPad Prism 9.0 software. For each parameter of all data presented, NS (No Significance), $*p < 0.05$, $** p< 0.01$, $*** p< 0.001$ and $**** p< 0.0001$. $p< 0.05$ is considered significant.

## Materials availability statement

Further information and requests for resources should be directed to and will be fulfilled by the lead contact, Zhonghan Yang (yangzhh@mail.sysu.edu.cn).

## Acknowledgements

The authors are grateful for the GTEx Portal database, human protein atlas database, Gene Expression Omnibus and PRESTO-Salsa database. This research was funded by the Shenzhen Science and Technology Project (grant number: JCYJ20190807154205627), Fund of Shenzhen Key Laboratory of Systems Medicine for inflammatory diseases (grant number: ZDSYS20220606100803007) received by Zhonghan Yang. The funders had no role in study design, data collection and analysis, decision to publish, or preparation of the manuscript.

## Additional information

### Funding

| Funder | Grant reference number | Author |
|---|---|---|
| Shenzhen Science and Technology Innovation Program | JCYJ20190807154205627 | Zhonghan Yang |
| Shenzhen Key Laboratory Fund | ZDSYS20220606100803007 | Zhonghan Yang |

The funders had no role in study design, data collection and interpretation, or the decision to submit the work for publication.

### Author contributions

Zewei Zhao, Conceptualization, Data curation, Formal analysis, Investigation, Writing – original draft, Writing – review and editing; Longyun Hu, Bigui Song, Tao Jiang, Qian Wu, Jiejing Lin, Xiaoxiao Li, Yi Cai, Jin Li, Bingxiu Qian, Siqi Liu, Data curation, Formal analysis; Jilu Lang, Conceptualization, Supervision, Writing – review and editing; Zhonghan Yang, Conceptualization, Supervision, Funding acquisition, Writing – review and editing

## Author ORCIDs
Zewei Zhao (iD) https://orcid.org/0000-0001-6731-0474
Zhonghan Yang (iD) https://orcid.org/0000-0002-3009-4074

## Ethics

All procedures related to animal feeding, treatment and welfare were conducted at Sun Yat-sen University Laboratory Animal Center. All the animal experiments were conducted with the approval of the Animal Care and Use Committee of Sun Yat-sen University (Approval ID: SYSU-IACUC-MED-2023-B082). This study was conducted in accordance with the ethical principles derived from the Declaration of Helsinki and Belmont Report and was approved by the review board of Sun Yat-sen University (Guangzhou, China).

Joint Public Review: https://doi.org/10.7554/eLife.100205.4.sa1
Author response https://doi.org/10.7554/eLife.100205.4.sa2

---

# Additional files

## Supplementary files

• Supplementary file 1. Primer sequences used for qPCR.

• Supplementary file 2. Gene annotation for screened genes. The 1134 screened genes were annotated by David database and 27 of these genes were identified as G-protein coupled receptor encoding gene.

• MDAR checklist

## Data availability

The transcriptomic dataset analyzed in this study can be accessed on the GTEx Portal database (https://gtexportal.org/home/multiGeneQueryPage), human protein atlas database (https://www.proteinatlas.org/) and GEO repository under accession number GSE118849. The PRESTO-Salsa dataset of ADGRA3 in this study can be accessed on the PRESTO-Salsa database (https://palmlab.shinyapps.io/presto-salsa/) (*Chen et al., 2023*). All other data generated or analyzed during this study are included in the manuscript and supporting files.

The following previously published datasets were used:

| Author(s) | Year | Dataset title | Dataset URL | Database and Identifier |
|---|---|---|---|---|
| Yu P, Li J, Deng SP, Zhang F | 2020 | Integrated analysis of a compendium of RNA-Seq data reveals putative master splicing factors underlying Rett syndrome and thermogenesis | https://www.ncbi.nlm.nih.gov/geo/query/acc.cgi?acc=GSE118849 | NCBI Gene Expression Omnibus, GSE118849 |

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
