## [Editor Report · eLife Assessment]

The study highlights adhesion G-protein-coupled receptor A3 (ADGRA3) as a potential target for activating adaptive thermogenesis in both white and brown adipose tissue. This finding offers **valuable** insights for researchers in the field of adipose tissue biology and metabolism. The authors have presented additional evidence to address the reviewers' comments, including experiments conducted on primary stromal vascular fractions from adipose tissues. However, the revised manuscript fails to address several reviewer concerns, such as the measurement of whole-body energy expenditure through indirect calorimetry and the assessment of food intake. Furthermore, the nanoparticle-mediated knockdown of Adgra3 did not adequately address the tissue selectivity of ADGRA in mice. As a result, the primary claims of the study are only partially supported by the available data, leading to the conclusion that the research is deemed **incomplete**.

---

## [Referee Report · Joint Public Review]

Based on bioinformatics and expression analysis using mouse and human samples, the authors claim that the adhesion G-protein coupled receptor ADGRA3 may be a valuable target for increasing thermogenic activity and metabolic health. Genetic approaches to deplete ADGRA3 expression in vitro resulted in reduced expression of thermogenic genes including Ucp1, reduced basal respiration and metabolic activity as reflected by reduced glucose uptake and triglyceride accumulation. In line, nanoparticle delivery of shAdgra3 constructs is associated with increased body weight, reduced thermogenic gene expression in white and brown adipose tissue (WAT, BAT), and impaired glucose and insulin tolerance. On the other hand, ADGRA3 overexpression is associated with an improved metabolic profile in vitro and in vivo, which can be explained by increasing the activity of the well-established Gs-PKA-CREB axis. Notably, a computational screen suggested that ADGRA3 is activated by hesperetin. This metabolite is a derivative of the major citrus flavonoid hesperidin and has been described to promote metabolic health. Using appropriate in vitro and in vivo studies, the authors show that hesperitin supplementation is associated with increased thermogenesis, UCP1 levels in WAT and BAT, and improved glucose tolerance, an effect that was attenuated in the absence of ADGRA3 expression.

The revised manuscript fails to address several reviewer concerns, such as the measurement of whole-body energy expenditure through indirect calorimetry and the assessment of food intake.

The previous reviews are here: https://elifesciences.org/reviewed-preprints/100205v2/reviews#tab-content

---

## [Author Response]

The following is the authors’ response to the previous reviews.

**Public Reviews:**

**Reviewer #1 (Public Review):**
Summary:This article identifies ADGR3 as a candidate GPCR for mediating beige fat development. The authors use human expression data from Human Protein Atlas and Gtex databases and combine this with experiments performed in mice and a murine cell line. They refer to a GPCR bioactivity screening tool PRESTO-Salsa, with which it was found that Hesperetin activates ADGR3. From their experiments, authors conclude that Hesperetin activates ADGR3, inducing a Gs-PKA-CREB axis resulting in adipose thermogenesis.Strengths:The authors analyze human data from public databases and perform functional studies in mouse models. They identify a new GPCR with a role in thermogenic activation of adipocytes.Considerations:Selection of ADGRA3 as a candidate GPCR relevant for mediating beiging in humans:The authors identify GPCRs that are expressed more highly in murine iBAT compared to iWAT in response to cold and assess which of these GPCRs are expressed in human subcutaneous or visceral adipocytes. Although this strategy will identify GPCRs that are expressed at higher levels in brown fat compared to beige and thus possibly more active in thermogenic function, the relevance in choosing GPCRs that also are expressed in unstimulated human white adipocytes should be considered. Thermogenic activity is not normally present in human white adipocytes. It would have strengthened the GPCR selection if the authors instead had assessed the intersection with human brown adipocytes that were activated with norepinephrine.

We appreciate your constructive feedback and believe that by adopting this refined strategy, we will strengthen our selection of GPCRs related to adipose thermogenesis in other ongoing studies. We look forward to continuing our research in this area and contributing to the understanding of adipose thermogenesis and its potential therapeutic applications. Thank you once again for your valuable input.

Strategy to investigate the role of ADGRA3 in WAT beiging:Having identified ADGRA3 as their candidate receptor, the authors investigated the receptor in mouse models, the murine inguinal adipocyte cell line 3T3 and in human subcutaneous adipose progenitors (HAdsc) differentiated in vitro. Calling the human cells "beige" is a stretch as these cells are derived from a white adipose depot. The authors do observe regulation in UCP1 and abundance of mitochondria following modification of ADGRA3 in the cells. However, in future studies, it should be considered if the receptor rather plays a role in differentiation per se, and perhaps not specifically in thermogenic differentiation/activity.

Regarding the reviewer's suggestion to consider whether ADGRA3 plays a role in differentiation per se, rather than specifically in thermogenic differentiation/activity, we acknowledge that this is an important consideration. Our current studies have focused on the role of ADGRA3 in regulating UCP1 expression and mitochondrial abundance, which are hallmarks of adipose thermogenic activity. However, we recognize that ADGRA3 may also have broader roles in adipocyte differentiation and function that are not limited to thermogenesis.

To address this point, in future studies, we plan to conduct additional experiments to investigate the potential role of ADGRA3 in adipocyte differentiation, including its effects on the expression of markers of adipocyte differentiation and its impact on adipocyte metabolism and function. These studies will provide further insights into the mechanisms by which ADGRA3 regulates adipocyte biology.

According to the Human Protein Atlas and Gtex databases, ADGRA3 is not only expressed in adipocytes, but also in other tissues and cell types. The authors address this by measuring the expression in a panel of these tissues, demonstrating a knockdown not only in the adipose tissue, but also in the liver and less pronounced in the muscle (Figure S2). It should thus be emphasized that the decreased TG levels in serum and liver in the mice might in fact depend on Adgra3 overexpression in the liver. Even though this might not have been the purpose of the experiment, it is important to highlight this as it could serve as hypothesis building for future studies of the function of this receptor.

Thank you for your thoughtful comments and feedback. We appreciate the insight provided by the Human Protein Atlas and Gtex databases regarding the tissue distribution of ADGRA3. We fully acknowledge that the decreased TG levels observed in both the serum and liver of the mice might be linked to the overexpression of *Adgra3* in the liver.

Although this was not the primary objective of our experiment, we agree that this observation is worth highlighting as it could serve as a basis for future hypothesis-driven research on the functional role of ADGRA3 in different tissues. In light of your comments, we emphasized this potential link between *Adgra3* overexpression in the liver and reduced TG levels in discussion, as follows.

“…the precise mechanisms underlying the influence of on adipose thermogenesis. Furthermore, it is crucial to highlight that the observed decrease in TG levels in both serum and liver (Figure 4-figure supplement 2C-D) might be attributed to the significant increase in *Adgra3* expression in the liver, which is a consequence of the nanoparticle-mediated overexpression of *Adgra3*. While the exact mechanism remains to be fully elucidated, this correlation suggests a potential link between *Adgra3* overexpression in the liver and reduced TG levels in the serum. We will employ more sophisticated models in subsequent studies to further…”

**Reviewer #3 (Public Review):**
Summary:The manuscript by Zhao et al. explored the function of adhesion G protein-coupled receptor A3 (ADGRA3) in thermogenic fat biology.Strengths:Through both in vivo and in vitro studies, the authors found that the gain function of ADGRA3 leads to browning of white fat and ameliorates insulin resistance.Weaknesses:There are several lines of weak methodologies such as using 3T3-L1 adipocytes and intraperitoneal(i.p.) injection of virus. Moreover, as the authors stated that ADGRA3 is constitutively active, how could the authors then identify a chemical ligand?Comments on revised version:The revised manuscript by Zhao et al. has limited improvement. The authors refused to perform revised experiments using primary cultures even though two reviewers pointed out the same weakness (3T3-L1 adipocytes are unsuitable). Using infrared thermography to measure body temperature is also problematic.

Thanks for your comments. We regret that human adipocytes induced from human adipose-derived stem cells (hADSCs) were not recognized as primary cultures by multiple reviewers. Therefore, we have included relevant experimental results of mouse primary adipocytes induced from stromal vascular fraction (SVF) in Figure 8E-H as a supplement. The thermal imaging device was used to measure the temperature of BAT, while the body temperature was measured at 9:00 using a rectal probe connected to a digital thermometer.